# Augmin prevents merotelic attachments by promoting proper arrangement of bridging and kinetochore fibers

**Valentina Štimac[†], Isabella Koprivec[†], Martina Manenica, Juraj Simunić, Iva M Tolić\***

Division of Molecular Biology, Ruđer Bošković Institute, Zagreb, Croatia

**Abstract** The human mitotic spindle is made of microtubules nucleated at centrosomes, at kinetochores, and from pre-existing microtubules by the augmin complex. However, it is unknown how the augmin-mediated nucleation affects distinct microtubule classes and thereby mitotic fidelity. Here, we use superresolution microscopy to analyze the previously indistinguishable microtubule arrangements within the crowded metaphase plate area and demonstrate that augmin is vital for the formation of uniformly arranged parallel units consisting of sister kinetochore fibers connected by a bridging fiber. This ordered geometry helps both prevent and resolve merotelic attachments. Whereas augmin-nucleated bridging fibers prevent merotelic attachments by creating a nearly parallel and highly bundled microtubule arrangement unfavorable for creating additional attachments, augmin-nucleated k-fibers produce robust force required to resolve errors during anaphase. STED microscopy revealed that bridging fibers were impaired twice as much as k-fibers following augmin depletion. The complete absence of bridging fibers from a significant portion of kinetochore pairs, especially in the inner part of the spindle, resulted in the specific reduction of the interkinetochore distance. Taken together, we propose a model where augmin promotes mitotic fidelity by generating assemblies consisting of bridging and kinetochore fibers that align sister kinetochores to face opposite poles, thereby preventing erroneous attachments.

**\*For correspondence:**
tolic@irb.hr

[†]These authors contributed equally to this work

**Competing interest:** The authors declare that no competing interests exist.

## Editor's evaluation

This manuscript investigates how the protein augmin contributes to correct spindle architecture and chromosome segregation during cell division in cultured human cells. High-quality super-resolution imaging and functional studies provide new insight into the mechanism of proper chromosome attachment to the spindle microtubules. This is an important paper that will be of interest to researchers interested in cell biology and cell biophysics.

## Introduction

The mitotic spindle has a sophisticated architecture that enables it to accurately segregate chromosomes during cell division. It consists of three major classes of microtubules: kinetochore microtubules that form kinetochore fibers (k-fibers) connecting chromosomes to the spindle pole through kinetochores, midplane-crossing microtubules that form antiparallel arrays in the central part of the spindle, and astral microtubules that extend from the spindle poles towards the cell cortex (**McIntosh, 2016**; **O'Toole et al., 2020**; **Prosser and Pelletier, 2017**). During metaphase and early anaphase, the majority of midplane-crossing microtubule bundles are laterally attached to a pair of sister k-fibers resembling a bridge between them, which is why they are called bridging fibers (**Kajtez et al., 2016**; **Vukušić et al., 2017**). These fibers balance the tension between sister kinetochores and maintain the

curved shape of the metaphase spindle (*Kajtez et al., 2016*; *Polak et al., 2017*; *Tolić and Pavin, 2016*). In addition to linking sister k-fibers, some midplane-crossing microtubules can also form connections between neighboring k-fibers (*O'Toole et al., 2020*).

Spindle microtubules in human somatic cells are generated by several nucleation mechanisms, including centrosome-dependent and augmin-dependent nucleation (*Kirschner and Mitchison, 1986*; *Pavin and Tolić, 2016*; *Petry, 2016*; *Prosser and Pelletier, 2017*; *Wu et al., 2008*; *Zhu et al., 2008*), with an addition of chromatin- and kinetochore-dependent nucleation as a third mechanism that contributes to the directional formation of k-fibers (*Maiato et al., 2004*; *Sikirzhytski et al., 2018*; *Tulu et al., 2006*). Centrosome-dependent nucleation was long thought to be predominant in spindle assembly; however, numerous studies revealed that a significant number of microtubules also arise from pre-existing microtubules, through augmin, an eight-subunit protein complex that serves as a recruiter of the γ-tubulin ring complex (γ-TuRC) required for microtubule nucleation (*David et al., 2019*; *Goshima et al., 2008*; *Kamasaki et al., 2013*; *Lawo et al., 2009*; *Song et al., 2018*; *Uehara et al., 2009*). Augmin-nucleated microtubules grow at an angle of 0-30° relative to the pre-existing microtubule (*Kamasaki et al., 2013*; *Petry et al., 2013*; *Verma and Maresca, 2019*) and show a directional bias towards kinetochores, resulting in the preserved polarity of the spindle once the initial kinetochore-microtubule attachments form (*David et al., 2019*; *Kamasaki et al., 2013*). Depletion of augmin complex in different cell types results in impairment of microtubule bundles within the spindle accompanied by the formation of long, curved bundles on the spindle periphery, loss of spindle bipolarity, shorter interkinetochore distance, chromosome misalignment, mitotic delays, and a higher incidence of aneuploidy and cytokinesis failure (*Almeida et al., 2022*; *Hayward et al., 2014*; *Uehara and Goshima, 2010*; *Uehara et al., 2009*; *Wu et al., 2008*; *Zhu et al., 2008*). Of the eight subunits in the complex, the two directly interacting subunits HAUS6 (hDgt6/FAM29A) and HAUS8 (hDgt4/Hice1) have been extensively studied because of their ability to interact with a γ-TuRC adapter protein NEDD1 and pre-existing microtubules, respectively (*Song et al., 2018*; *Uehara et al., 2009*). While previous studies mainly focused on the effect of augmin on astral and kinetochore microtubules, the effect on midplane-crossing microtubules remains largely unexplored (*Almeida et al., 2022*; *Hayward et al., 2014*; *Song et al., 2018*; *Uehara et al., 2009*; *Uehara et al., 2016*; *Uehara and Goshima, 2010*; *Wu et al., 2008*; *Zhu et al., 2008*). Recent electron tomography work on spindles in human cells showed that ends of midplane-crossing microtubules interact with the wall of kinetochore microtubules (*O'Toole et al., 2020*), indicating that augmin-dependent nucleation might play an important role in their formation.

Augmin depletion has previously been linked to higher incidence of segregation errors (*Wu et al., 2008*) and the appearance of lagging chromosomes (*Almeida et al., 2022*; *Viais et al., 2021*), which were connected to impaired brain development in a recent study (*Viais et al., 2021*). Homozygous loss of HAUS6 subunit of the augmin complex was also seen in several cancer types, such as sarcomas, pancreatic adenocarcinomas, gliomas, and glioblastomas (ICGC/TCGA, 2020, retrieved by using cBio-Portal *Cerami et al., 2012*; *Gao et al., 2013*). However, the origin of segregation errors in augmin depletion remains largely unexplored due to extensive mitotic delays often experienced by these cells (*Wu et al., 2008*).

To explore how the augmin-dependent microtubule nucleation affects functionally distinct microtubule bundles and thereby mitotic fidelity, we depleted augmin in hTERT-RPE1 and HeLa cells and imaged them using stimulated emission depletion (STED) (*Hell and Wichmann, 1994*; *Klar and Hell, 1999*) and confocal microscopy. We show that the augmin complex plays a vital role in the formation of highly organized microtubules required for mitotic fidelity. Augmin depletion leads to a three-fold increase in segregation errors when the checkpoint is weakened. A significant number of lagging chromosomes following augmin depletion have a reduced tension and a large tilt with respect to the spindle axis in metaphase, which may facilitate the formation of merotelic attachments within the disorganized spindle area of augmin-depleted cells. The appearance of severely disordered microtubules occurs along with the strong reduction in the number of proper bridging microtubules connecting sister k-fibers. Interestingly, the interkinetochore distance after augmin depletion was larger for kinetochore pairs with bridging fibers than for those without, indicating a specific effect of the augmin-generated bridging fibers on interkinetochore tension. Taken together, we propose that augmin affects mitotic fidelity by forming highly organized microtubule arrangements consisting of two sister k-fibers connected by a bridging fiber. In these arrangements oriented parallel to the

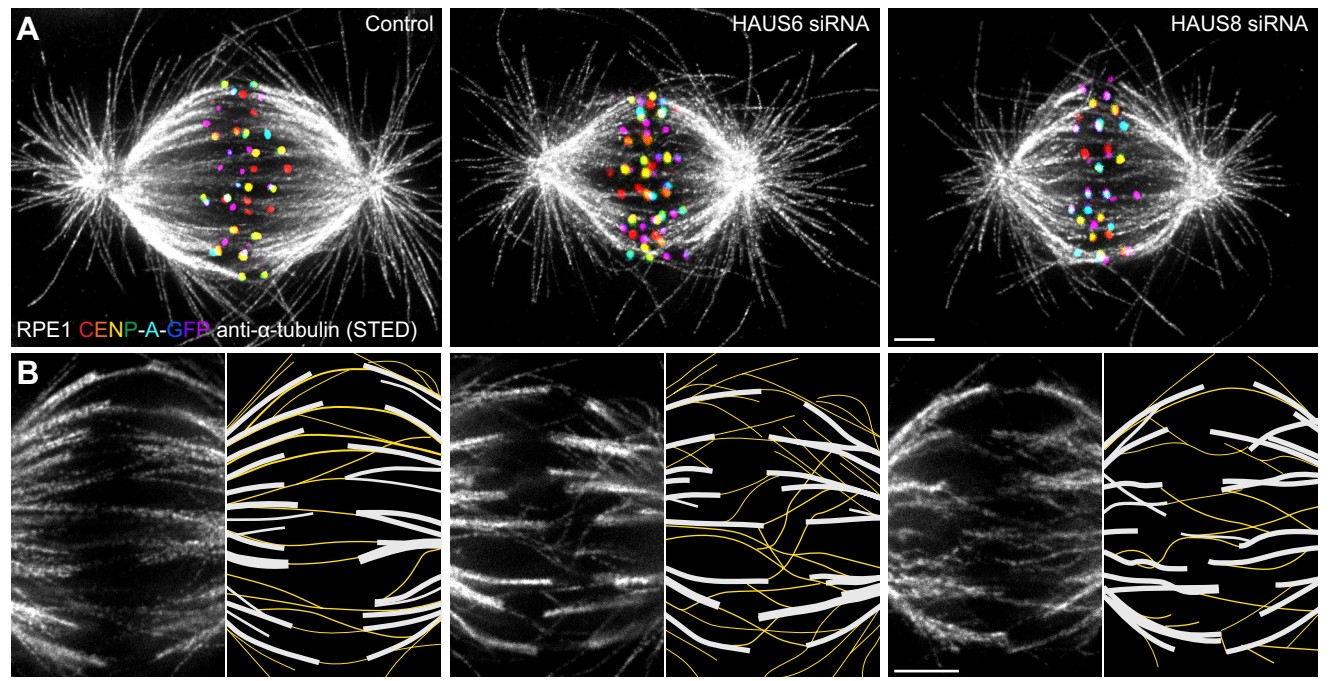

**Figure 1.** Augmin ensures the proper formation of entities consisting of bridging fibers that connect two sister k-fibers. (**A**) STED superresolution images of microtubules immunostained for α-tubulin (gray) in control (left), HAUS6- (middle), and HAUS8-depleted (right) RPE1 cells stably expressing CENP-A-GFP (rainbow, confocal). Images show maximum intensity projections of 6 central z-planes of metaphase spindles. Kinetochores are color-coded for depth from blue to red with the Spectrum LUT in ImageJ. (**B**) Insets of STED superresolution images of microtubules (gray) in spindle midzones of control (left), HAUS6- (middle) and HAUS8-depleted (right) cells. Next to each image is the schematic representation of the microtubules in the midzone, with white lines representing k-fiber microtubules and yellow lines representing midplane-crossing microtubules. Images show a single z-plane and do not correspond to midzones of spindles in panel (**A**). All images are adjusted for clarity based on the intensity of astral microtubules in each image (see Materials and methods). Scale bars, 2 μm.

The online version of this article includes the following figure supplement(s) for figure 1:

**Figure supplement 1.** Augmin is necessary for the formation of uniformly arranged microtubule bundles in the spindle.

spindle axis, augmin-nucleated bridging fibers prevent erroneous kinetochore-microtubule attachments during metaphase, while augmin-nucleated k-fibers resolve them in anaphase.

## Results

### Augmin is vital for the formation of uniformly arranged units consisting of two sister k-fibers connected by a bridging fiber

To overcome the limitations of confocal microscopy and explore the relationship between different classes of microtubules within the crowded metaphase plate area, we performed stimulated emission depletion (STED) superresolution imaging (*Hell and Wichmann, 1994*; *Klar and Hell, 1999*) of microtubules within the bipolar metaphase spindles of human cells (*Figure 1A–B*, *Figure 1—figure supplement 1*). This enabled us to easily distinguish between k-fibers that start at kinetochores and midplane-crossing microtubules that pass through the central part of the spindle. HAUS6 or HAUS8 components of the augmin complex were depleted by siRNA in hTERT-RPE1 (hereafter referred to as RPE1) cells stably expressing CENP-A-GFP and immunostained for tubulin (*Figure 1A*, *Figure 1—figure supplement 1*). Augmin depletion was confirmed using immunocytochemistry and western blot analysis (*Figure 2—figure supplement 1A-CFigure 2—source data 1*).

In control cells, the vast majority of midplane-crossing microtubule bundles laterally attached to a pair of sister k-fibers and formed a bridging fiber between them, consistent with previous findings (*Kajtez et al., 2016*; *Vukušić et al., 2017*). These bridging fibers were nearly-parallel with respect to one another and the spindle axis. Additionally, a small portion of midplane-crossing microtubules

formed a secondary connection between a k-fiber on one side and a non-sister k-fiber on the other side (*O'Toole et al., 2020*). In comparison with untreated cells, midplane-crossing microtubules after HAUS6 or HAUS8 depletion extended at a variety of angles, and were wavy and disordered, particularly in the inner part of the spindle close to the pole-to-pole axis. Strikingly, midplane-crossing microtubules less often formed bridging fibers that connect to sister k-fibers in cells depleted of HAUS6 or HAUS8. Instead, they formed more complex arrangements, primarily consisting of one or more connections between various k-fibers within the metaphase spindle. This was contrary to k-fibers which, even though often missing a bridge between them, appeared relatively similar to those in control cells (*Figure 1B*, *Figure 1—figure supplement 1*). Taken together, augmin is vital for the proper organization of midplane-crossing microtubules into uniformly arranged bridging fibers that connect two sister k-fibers and extend nearly-parallel to the spindle axis.

## Augmin helps both prevent and resolve segregation errors through joint action of bridging and k-fibers

The appearance of disordered midplane-crossing microtubules in the metaphase plate area of spindles without augmin prompted us to investigate whether these microtubules affect mitotic fidelity. Augmin depletion has previously been linked to higher incidence of segregation errors (*Almeida et al., 2022*; *Viais et al., 2021*; *Wu et al., 2008*), but their origin remained largely unexplored due to extensive mitotic delays in augmin-depleted cells (*Wu et al., 2008*). To avoid mitotic delays, we performed live-cell confocal imaging for which we codepleted the checkpoint protein Mad2 together with HAUS6 to induce anaphase onset, and used Mad2-depleted cells (*Mayr et al., 2007*), which have only a few segregation errors, as a control (*Figure 2A*, *Videos 1 and 2*). To explore the mechanistic origin of segregation errors, we divided them into three distinct groups: misaligned chromosomes in which both kinetochores were found outside the metaphase plate just before anaphase onset, lagging chromosomes in which the kinetochore is visibly stretched and positioned in the central part of the spindle while other kinetochores are already separating, and other less common and diverse errors (*Figure 2A–B*, See Materials and methods).

The treatment with Mad2 siRNA resulted in a total of 1.5±0.2 segregation errors per cell (all data are given as mean ± SEM). However, the effect was significantly more severe when Mad2 was codepleted with HAUS6, resulting in a total of 5.7±0.8 segregation errors per cell (*Figure 2B*). Tracking of sister kinetochores in live-cell videos revealed that the number of misaligned kinetochore pairs per cell was similar in both Mad2 depletion and Mad2/HAUS6 codepletion (*Figure 2B*), which is why we presume they appeared due to Mad2 depletion independently of HAUS6 depletion. Nevertheless, they were much more likely to missegregate in cells without HAUS6, where 80 ± 9% of kinetochore pairs jointly segregated into the same cell, compared to only 20 ± 13% in control cells (*Figure 2B*). Interestingly, lagging kinetochores were both more frequent and more likely to missegregate in cells with Mad2/HAUS6 codepletion, in which there were 2.6±0.4 lagging kinetochores per cell, and 45 ± 6% of all lagging kinetochores ultimately missegregated. In contrast, there were only 0.8±0.2 lagging kinetochore pairs per cell in Mad2 depletion, and 77 ± 8% of them segregated correctly (*Figure 2B*). Finally, errors classified as others were more frequent in cells with Mad2/HAUS6 codepletion when compared to those with Mad2 depletion, but not more prone to missegregation (*Figure 2B*).

As misaligned chromosomes appeared equally frequently in Mad2 depletion and Mad2/HAUS6 codepletion, but differed in their ability to correctly segregate, we used them as a tool to isolate the role of augmin in resolving segregation errors during anaphase. Consistent with previous findings (*Uehara et al., 2009*), both poleward movement of kinetochores during anaphase A and spindle elongation during anaphase B were reduced following Mad2/HAUS6 codepletion (*Figure 2—figure supplement 1D-F*). To analyze in detail the movement of misaligned kinetochore pairs, we tracked both kinetochores with respect to the proximal and distal pole, that is the pole which is closer to the misaligned kinetochore pair or the pole which is further away from it, respectively. The kinetochore closer to the proximal pole approached the proximal pole during anaphase A both after Mad2 depletion and Mad2/HAUS6 codepletion, thereby moving towards the pole to which it should segregate in both cases (*Figure 2C*). However, the kinetochore further away from the proximal pole usually remained stagnant for a short period of time, and afterwards typically moved towards the distal pole and accurately segregated as a 'lazy' kinetochore (*Sen et al., 2021*) in Mad2-depleted cells. In contrast, the kinetochore further away from the proximal pole experienced a short stagnation period

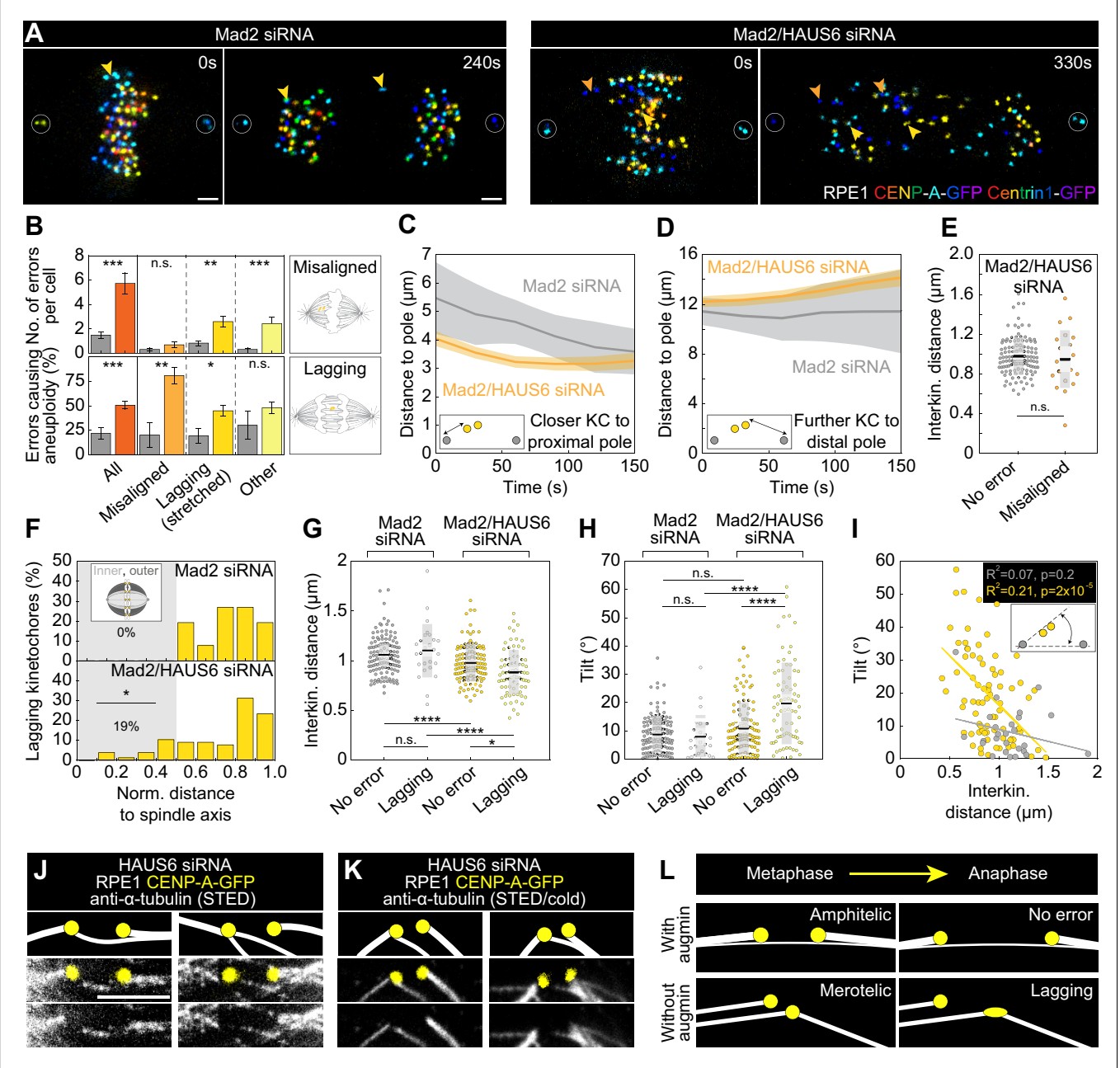

**Figure 2.** Augmin-nucleated midplane-crossing microtubules prevent kinetochore tilt and thus merotelic attachments. (**A**) Time-lapse images of RPE1 cells stably expressing CENP-A-GFP and Centrin1-GFP (rainbow, confocal) in Mad2-depleted cells (left) and Mad2/HAUS6-codepleted cells (right). Yellow arrows represent lagging kinetochores and orange arrows misaligned kinetochores. Kinetochores are color-coded for depth from blue to red with the 16 Colors LUT in ImageJ. (**B**) The number of segregation errors per cell (top) and the percentage of errors causing aneuploidy (bottom) in Mad2-depleted cells (gray) and Mad2/HAUS6-codepleted cells (dark orange, light orange, dark yellow, light yellow). All segregation errors (dark orange) are divided into three groups: misaligned (light orange), lagging (dark yellow) and other (light yellow). Schematic representations next to the graph represent misaligned kinetochores (top) and lagging kinetochore (bottom). The number of errors in Mad2-depleted cells - in total 46 errors in 22 out of 31 cells; 10 misaligned kinetochore pairs in 9 out of 31 cells; 26 lagging kinetochores in 16 out of 31 cells; 10 other errors in 9 out of 31 cells. Number of errors in Mad2/HAUS6-codepleted cells - in total 172 errors in 25 out of 30 cells; 21 misaligned kinetochore pairs in 11 out of 30 cells; 78 lagging kinetochores in 23 out of 30 cells; 73 other errors in 20 out of 30 cells. Aneuploidy in Mad2-depleted cells - in total 11/46 errors in 22 out of 31 cells; 2/10 misaligned kinetochore pairs in 9 out of 31 cells; 6/26 lagging kinetochores in 16 out of 31 cells; 3/10 other errors in 9 out of 31 cells. Aneuploidy in Mad2/HAUS6-codepleted cells - in total 47/172 errors in 25 out of 30 cells; 17/21 misaligned kinetochore pairs in 11 out of 30 cells; 35/78 lagging kinetochores in 23 out of 30 cells; 35/73 other errors in 20 out of 30 cells. (**C**) The distance of closer kinetochore to the proximal pole and (**D**) further kinetochore to the distal pole of misaligned kinetochore pairs in time for Mad2-depleted (gray) and Mad2/HAUS6-codepleted cells (orange). Values

*Figure 2 continued on next page*

*Figure 2 continued*

are shown as mean (dark line) and SEM (shaded areas). The insets show the positions of kinetochores (yellow) with respect to spindle poles (gray). (**E**) Univariate scatter plot of the interkinetochore distance of error-free kinetochore pairs (gray) and misaligned kinetochore pairs (orange) in Mad2/HAUS6-codepleted cells. N=30 cells and 120 error-free kinetochore pairs in Mad2-depleted cells and N=30 cells and 21 misaligned kinetochore pairs in Mad2/HAUS6-codepleted cells. (**F**) The percentage of lagging kinetochores in Mad2-depleted (top) and Mad2/HAUS6-codepleted cells (bottom) divided by their location with respect to the pole-to-pole axis into inner and outer (schematic representation shown as inset, see Materials and methods). (**G**) Univariate scatter plot of the interkinetochore distance of error-free and lagging kinetochore pairs in Mad2-depleted (gray) and Mad2/HAUS6-codepleted cells (yellow). (**H**) Univariate scatter plot of the angle that the error-free and lagging kinetochore pairs form with the pole-to-pole axis (tilt) in Mad2-depleted (gray) and Mad2/HAUS6-codepleted cells (yellow). N=31 cells and 124 error-free and 26 lagging kinetochore pairs from Mad2-depleted cells. N=30 cells and 120 error-free and 78 lagging kinetochore pairs from Mad2/HAUS6-codepleted cells. (**I**) The correlation of the tilt and the interkinetochore distance for Mad2-depleted (gray) and Mad2/HAUS6-codepleted cells (yellow). Inset shows schematic representation of the tilt (kinetochores are shown in yellow and spindle poles in gray). (**J**) The insets of kinetochore pairs with merotelic attachments in RPE1 cells stably expressing CENP-A-GFP (yellow, confocal) and immunostained for α-tubulin (gray, STED). (**K**) The insets of kinetochore pairs from cells as in (**J**) but exposed to cold treatment. Images in (**J**) and (**K**) are smoothed with 0.5-mm-sigma Gaussian blur and adjusted for clarity (see Materials and methods). Schematic representations in (**J**) and (**K**) are shown above the insets for better visualization of merotelic microtubule attachments. (**L**) The schematic representations of a kinetochore pair with amphitelic attachment in metaphase that does not cause any segregation errors during anaphase when augmin is present (top) and a kinetochore pair with merotelic attachment in metaphase that ends up as the lagging kinetochore during anaphase when augmin is not present (bottom). (**E, G and H**) Boxes represent standard deviation (dark gray), 95% confidence interval of the mean (light gray) and mean value (black). All results were obtained from three independent experiments. Statistical analysis (B (top) and E) Mann–Whitney U test; (B (bottom) and F) Fisher's exact test; (**G and H**) ANOVA with the post-hoc Tukey test; (**I**) linear regression; p-value legend: <0.0001 (****), 0.0001–0.001 (***), 0.001–0.01 (**), 0.01–0.05 (*), ≥0.05 (ns). Scale bars, 2 µm.

The online version of this article includes the following source data and figure supplement(s) for figure 2:

**Source data 1.** Immunoblot analysis of HAUS6 siRNA treatment efficiency in RPE1 cells stably expressing CENP-A-GFP and Centrin1-GFP.

**Figure supplement 1.** Augmin is required for accurate chromosome segregation during anaphase.

and then typically moved away from the pole to which it should segregate, thereby missegregating in Mad2/HAUS6-codepleted cells (*Figure 2D*, *Figure 2—figure supplement 1G-H*). As the interkinetochore distance of both correctly and incorrectly segregating misaligned kinetochore pairs in Mad2/HAUS6-codepleted cells was similar (*Figure 2E*), the absence of biorientation is unlikely to be the cause of missegregation for these kinetochores. Instead, missegregation likely occurs due to k-fibers with fewer microtubules (*Almeida et al., 2022*; *Uehara et al., 2009*; *Zhu et al., 2008*) creating insufficient force to move the kinetochore towards the distal pole (*Dudka et al., 2018*), against the movement of neighboring kinetochores and the corresponding chromosome mass which travel towards the proximal pole (*Figure 2A*, *Video 2*).

Whereas lagging chromosomes have been previously observed following augmin depletion (*Almeida et al., 2022*; *Viais et al., 2021*; *Wu et al., 2008*), their origin remains unknown. Because we observed that disorganized midplane-crossing microtubules were often concentrated in the inner part of the mitotic spindle near the main spindle axis upon augmin depletion (*Figure 1B*, *Figure 1—figure supplement 1*), we decided to investigate the spatial distribution (*Figure 2F*) of lagging kinetochores in Mad2-depleted and Mad2/HAUS6-codepleted cells to see if their increased number might be connected to this phenotype. In Mad2-depleted

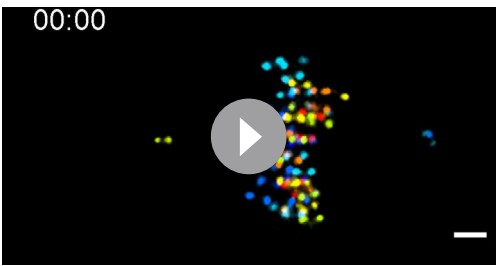

**Video 1.** RPE1 cell stably expressing CENP-A-GFP and Centrin1-GFP (16-colors) following Mad2 depletion. Kinetochores are color-coded for depth from blue to red with the 16 Colors LUT and noise was processed with the Despeckle function in ImageJ. Scale bar, 2 µm.
https://elifesciences.org/articles/83287/figures#video1

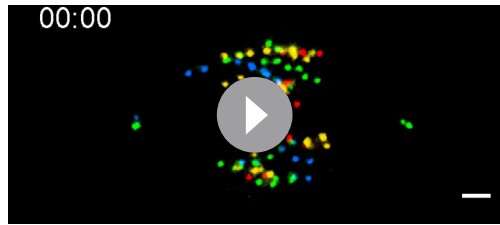

**Video 2.** RPE1 cell stably expressing CENP-A-GFP and Centrin1-GFP (16-colors) following Mad2/HAUS6 codepletion. Kinetochores are color-coded for depth from blue to red with the 16 Colors LUT and noise was processed with the Despeckle function in ImageJ. Scale bar, 2 µm.
https://elifesciences.org/articles/83287/figures#video2

cells, all lagging kinetochore pairs were situated in the outer half of the spindle just before anaphase onset. Remarkably, in Mad2/HAUS6-codepleted cells, 19 ± 5% of all lagging kinetochore pairs were situated in the inner part of the spindle just before anaphase onset (*Figure 2F*), where disordered midplane-crossing microtubules most frequently appeared. Thus, organization of midplane-crossing microtubules into bridging fibers might play an important role in mitotic fidelity.

To further explore how this compromised spindle geometry affects lagging kinetochores, we measured their interkinetochore distance just before anaphase onset. The lagging kinetochore pairs in Mad2/HAUS6-codepleted cells had an interkinetochore distance of 0.89±0.03 μm, which was significantly smaller than the interkinetochore distance of 0.98±0.02 μm measured in error-free kinetochore pairs (*Figure 2G*). This was not the case for lagging kinetochore pairs in Mad2-depleted cells which had an interkinetochore distance of 1.10±0.05 μm, similar to the interkinetochore distance of 1.05±0.02 μm measured in error-free kinetochore pairs (*Figure 2G*). Distinctly reduced interkinetochore distance of lagging kinetochore pairs following Mad2/HAUS6 codepletion suggests that they appear due to compromised spindle architecture being unable to maintain adequate kinetochore tension.

As sister k-fibers in the disorganized spindle region were sometimes diagonally positioned with respect to the pole-to-pole axis, we decided to test if this tilt is also connected to the appearance of segregation errors by measuring the angle that either lagging or error-free kinetochores form with the spindle axis just before anaphase onset. As for the interkinetochore distance following Mad2/HAUS6 codepletion, lagging kinetochore pairs were different from error-free kinetochore pairs, the tilt of which was 19.7±1.6° and 10.9±0.8°, respectively (*Figure 2H*). In contrast, in Mad2-depleted cells with preserved spindle geometry there was no difference between lagging kinetochore pairs with the tilt of 8.2±1.5° and error-free kinetochore pairs with the tilt of 8.9±0.6°, which was also similar to the tilt of 10.9±0.8° measured for error-free kinetochore pairs following Mad2/HAUS6 codepletion (*Figure 2H*). The tilt of kinetochores inversely correlated with the interkinetochore distance in augmin depletion, but not in control cells (*Figure 2I*). These data indicate the importance of nearly parallel configuration of kinetochore pairs during metaphase for mitotic fidelity, and points to the augmin-specific cause of lagging kinetochores that likely arise due to compromised and tilted bundle architecture facilitating the formation of merotelic attachments. Indeed, we found merotelic attachments in HAUS6-depleted cells imaged using STED microscopy, with most kinetochores forming an attachment with the microtubule from the opposite side of the mitotic spindle, while missing a proper bridging fiber (*Figure 2J*). To further test the nature of the observed attachments, we combined STED microscopy with cold treatment to remove midplane-crossing microtubules and preserve only kinetochore microtubules (*DeLuca et al., 2006*; *Sacristan et al., 2018*; *Silkworth et al., 2012*), which allowed us to confirm their true merotelic nature (*Figure 2K*). Interestingly, our live-cell imaging experiments reveal that augmin is required not only to prevent the formation of merotelic attachments in metaphase, but also to resolve them in anaphase, as a larger percentage of lagging kinetochore pairs ends up missegregating in Mad2/HAUS6 codepletion than in Mad2 depletion (*Figure 2B*). This suggests that the insufficient force provided by k-fibers with fewer microtubules (*Dudka et al., 2018*), which is responsible for missegregation of misaligned kinetochore pairs following augmin depletion, also leads to inability to resolve merotelic attachments during anaphase.

Altogether, we propose that augmin ensures mitotic fidelity through the joint action of bridging and k-fibers. While augmin-nucleated bridging fibers prevent merotelic attachments by creating a nearly parallel and highly bundled spindle geometry unfavorable for creating additional attachments, augmin-nucleated k-fibers produce robust force required to resolve any potentially appearing errors during anaphase (*Figure 2L*).

## Bridging fibers are predominantly generated through augmin-dependent nucleation

As our visual assessment revealed that spindles without augmin have disorganized arrangements of midplane-crossing microtubules and often lack proper bridging fibers (*Figure 1B*), which were also missing at kinetochore pairs that formed merotelic attachments (*Figure 2J*), we set out to analyze how augmin-dependent microtubule nucleation contributes to the formation of bridging fibers in immunostained RPE1 cells imaged using STED microscopy (*Figure 3A–B*). Bridging fibers were strictly defined as midplane-crossing microtubules that connect two sister k-fibers, whereas k-fibers were

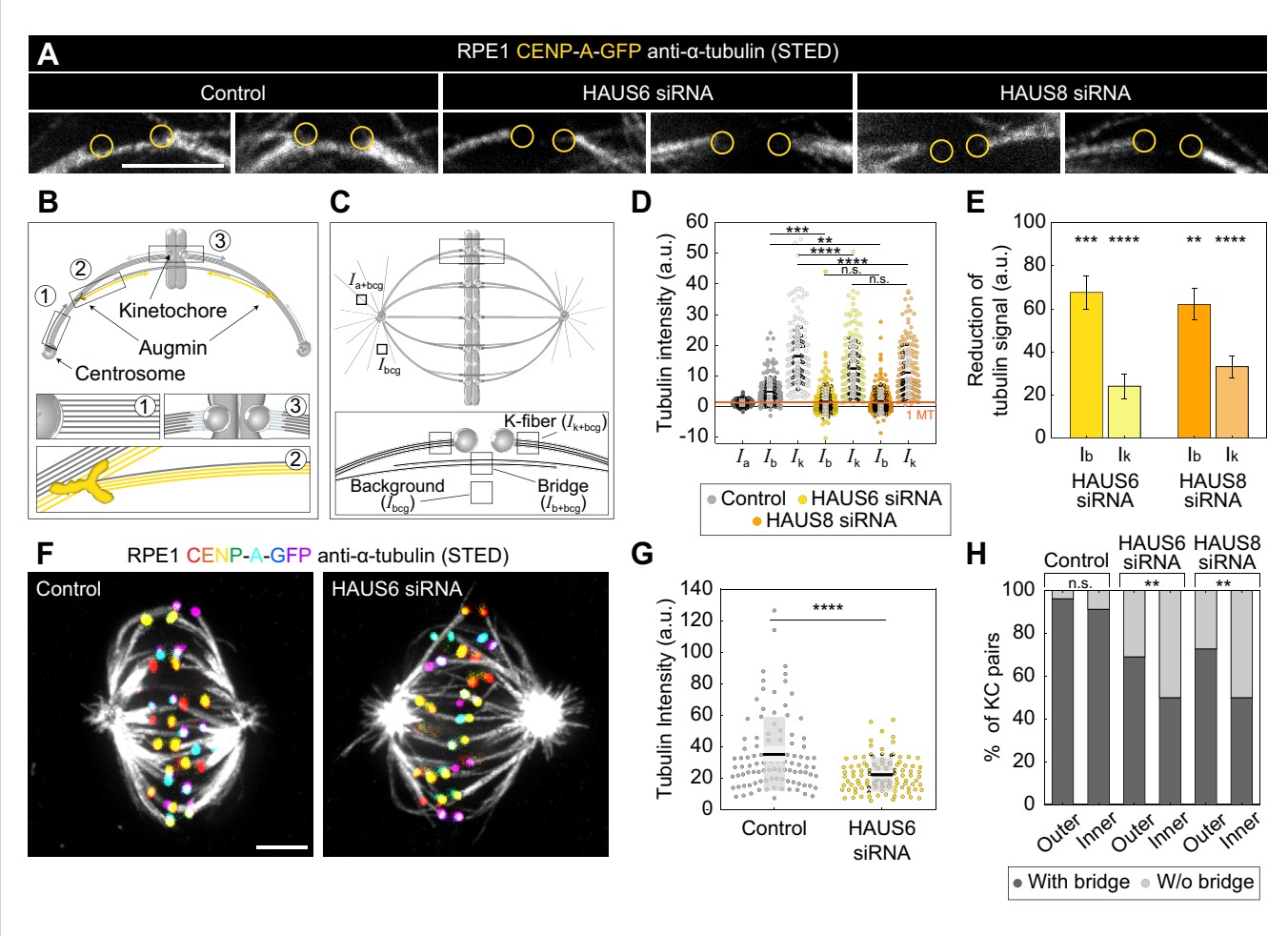

**Figure 3.** Augmin is crucial for the nucleation of bridging microtubules. (**A**) The insets of kinetochore pairs in RPE1 cells stably expressing CENP-A-GFP (not shown) immunostained for α-tubulin (gray, STED) in control cells (left) and after HAUS6 (middle) or HAUS8 (right) depletion. The insets demonstrate kinetochore pairs with bridging fibers affected by HAUS6 or HAUS8 depletion compared to bridging fibers in control cells. The positions of kinetochores are marked with yellow circles. (**B**) The schematic representation of three possible pathways of microtubule nucleation: (1) centrosome-dependent (2) augmin-dependent and (3) chromatin- and kinetochore-dependent nucleation. The augmin complex is shown in yellow. (**C**) Top: the schematic representation of the mitotic spindle in metaphase and the method used to measure the tubulin intensity of the astral microtubules. Small square regions were measured on microtubules extending from the spindle pole, corresponding to astral microtubules. Their background was measured in the empty area between the two astral microtubules, and it was subtracted from astral microtubule intensity. Bottom: Schematic representation of the method used to measure the tubulin intensity of the bridging and k-fiber. Small square regions were measured between two kinetochores or right next to the kinetochore, corresponding to bridging and k-fibers, respectively. The intensity of k-fibers was measured as an average of two sister k-fibers, and the average value of the background within the spindle was subtracted from all measurements. $I_{a+bcg}$ = intensity of astral microtubules with background, $I_{k+bcg}$ = intensity of k-fibers with background, $I_{b+bcg}$ = intensity of bridging microtubules with background, $I_{bcg}$ = intensity of background. (**D**) Univariate scatter plot of tubulin signal intensities of astral microtubules in control cells (reference value, dark gray, $I_a$), and bridging fibers ($I_b$) and k-fibers ($I_k$) in control cells (gray), HAUS6- (yellow) and HAUS8-depleted cells (orange). (**E**) The reduction of tubulin signal in the bridging fiber ($I_b$) and the k-fiber ($I_k$) following HAUS6 (yellow) or HAUS8 (orange) depletion, values are shown as mean ± SEM. p-Values were calculated using the absolute values of tubulin signal intensity of bridging or k-fibers following HAUS6 or HAUS8 depletion, compared to the absolute values of tubulin signal intensity of corresponding fibers in control cells. (**D**) and (**E**) N=30 cells and 90 astral microtubules in control cells, 158 bridging and sister k-fibers in control and 180 bridging and sister k-fibers in HAUS6- and HAUS8 siRNA-treated cells. (**F**) STED superresolution images of microtubules stained for α-tubulin (gray) in RPE1 cells stably expressing CENP-A-GFP (rainbow, confocal) in control cells (left) and HAUS6 siRNA-treated cells (right) exposed to cold treatment. The images are maximum intensity projections and kinetochores are color-coded for depth from blue to red with the Spectrum LUT in ImageJ. (**G**) Univariate scatter plot of the tubulin signal intensities of k-fibers in control cells (gray) and upon HAUS6 depletion (yellow) in cells exposed to cold treatment. N=30 cells and 101 bundles in control cells and 102 bundles in HAUS6-depleted cells. (**H**) The fractions of kinetochore pairs with bridging fibers (dark gray) and with undetectable bridging fibers (light gray) in control, HAUS6- and HAUS8-depleted cells. Kinetochore pairs are divided based on their location in the spindle into outer and inner (See Materials and methods and Results). (**D**) and (**G**) Boxes represent standard deviation (dark gray), 95% confidence interval of the mean (light gray) and mean value (black). All results were obtained from three independent experiments. Statistical analysis (**D**

*Figure 3 continued on next page*

*Figure 3 continued*

**and E**) ANOVA with post-hoc Tukey test, (**G**) Mann–Whitney U test, (**H**) chi-square test; p-value legend:<0.0001 (****), 0.0001–0.001 (***), 0.001–0.01 (**), 0.01–0.05 (*), ≥0.05 (ns). All images are adjusted for clarity (see Materials and methods). Scale bars, 2 μm.

The online version of this article includes the following figure supplement(s) for figure 3:

**Figure supplement 1.** Augmin ensures proper architecture and dynamics of the metaphase spindle.

defined as microtubules that start at kinetochores (*Figure 3C*). We measured tubulin signal intensity of randomly selected bridging ($I_b$) and k-fibers ($I_k$) which had no other microtubules in their immediate neighborhood, in a small square region between two kinetochores or at the pole-side of kinetochore, respectively (*Figure 3C*, see Materials and methods). By using the resulting tubulin signal intensities, we first estimated the number of microtubules in the bridging fiber in untreated RPE1 cells. Electron tomography of spindles in RPE1 cells showed that k-fibers consist of $n_k$ = 12.6 ± 1.7 microtubules (*O'Toole et al., 2020*). Thus, the bridging fiber consists of $n_b = I_b \times n_k / I_k$ = 3.8 ± 0.4 microtubules (for the explanation of $I_b$ and $I_k$ calculations, see Materials and methods). The accuracy of this calculation was additionally tested by measuring the intensity of astral microtubules, which presumably consist of single microtubules (*McDonald et al., 1992*). Indeed, using the number of microtubules in the k-fiber, our measurement of astral microtubule intensities showed that the astral microtubules consist of $n_a$ = 1.0 ± 0.1 microtubules (*Figure 3D*).

Quantification of STED images further revealed that HAUS6 depletion resulted in 68 ± 8% reduction of the bridging fiber signal intensity and 24 ± 6% reduction of the k-fiber signal intensity, with similar results obtained by HAUS8 depletion (*Figure 3D–E*). These data indicate that augmin depletion affects not only k-fibers, but even more so bridging fibers. The contribution of augmin to the nucleation of k-fibers was independently tested by measuring their intensity in spindles exposed to cold treatment in which bridging fibers are removed (*Figure 3F*). HAUS6 depletion resulted in a 37 ± 5% reduction of the k-fibers (*Figure 3G*), which is consistent with a previous study (*Zhu et al., 2008*) and comparable to values under non-cold conditions. Based on the measurements under non-cold conditions, we estimate that after HAUS6 depletion bridging fibers consist of 1.2±0.7 microtubules and k-fibers of 9.6±1.5 microtubules, which we interpret as microtubules nucleated in an augmin-independent manner. Thus, 2.6±0.7 microtubules in the bridging fiber and 3.0±0.9 microtubules in the k-fiber are nucleated in an augmin-dependent manner.

Remarkably, 41 ± 4% of all kinetochore pairs in HAUS6-depleted cells had no detectable bridging fibers, defined as those with the tubulin signal below the background signal (see Materials and methods), and consistent with results obtained using visual inspection (*Figure 3—figure supplement 1A*). The majority of kinetochore pairs without bridging fibers were located in the inner part of the mitotic spindle, where as much as 50 ± 5% of all kinetochore pairs had undetectable bridging fibers after augmin depletion, compared to only 27 ± 5% in the outer part (*Figure 3H*), thus pointing to an irregular and more complex spatial distribution of bridging fibers in the inner part of the spindles following augmin depletion. Similar results were obtained from superresolution imaging after HAUS8 depletion (*Figure 3H*). To further validate our results, we performed live-cell confocal imaging with SiR-tubulin (*Lukinavičius et al., 2014*) and analyzed the spindles by two independent methods (*Figure 3—figure supplement 1A–H*, See Materials and methods). Altogether, these results reveal that the augmin complex is a major nucleator of bridging fibers, whereas its contribution to the formation of k-fibers is significant but less prominent.

The compromised microtubule nucleation following augmin depletion led to the impairment of overall spindle geometry, creating a unique system where three main types of interactions between k-fibers and bridging fibers can be found within the same spindle: (1) sister k-fibers attached to bridging fibers, (2) sister k-fibers without a bridging fiber, and (3) solitary, long, interpolar bundles without associated kinetochores. This is in contrast with control cells, where the first group dominates and the other two groups are rarely found (*Polak et al., 2017*). To gain insight into the contribution of each of these functionally distinct microtubule bundles to the maintenance of spindle geometry, we traced the outermost bundles in HAUS6 siRNA-treated RPE1 cells imaged using STED microscopy and fitted a circle to the bundle outline (*Figure 3—figure supplement 1*, see Materials and methods). Whereas the bundles without kinetochores in HAUS6 siRNA-treated cells had a significantly longer contour when compared to all other bundle types (*Figure 3—figure supplement 1J*), k-fibers without bridging fibers in augmin-depleted cells had a significantly larger radius of curvature than any of the

other bundle types in augmin-depleted or control cells (*Figure 3—figure supplement 1K*). Taken together, the outer interpolar bundles without associated kinetochores are excessively long and make the spindle wider, whereas k-fibers lacking a bridging fiber are overly straight, ultimately resulting in a diamond-like shape of the spindle. This change in spindle shape in the absence of proper bridging fibers is consistent with the prediction of our theoretical model (*Kajtez et al., 2016*) and previous experiments (*Jagrić et al., 2021*).

In addition to spindle architecture, compromised microtubule nucleation following augmin depletion also affected spindle dynamics, as poleward flux in U2OS cells stably expressing CENP-A-GFP, mCherry-tubulin and photoactivatable-GFP-α-tubulin was significantly reduced (*Figure 3—figure supplement 1L-M*), in agreement with findings in Indian Muntjac cells (*Almeida et al., 2022*). Recent speckle microscopy experiments in RPE1 cells, which were able to separate the effect of augmin on poleward flux of bridging and k-fibers, revealed that both k-fibers and the remaining bridging fibers were significantly slowed down (*Risteski et al., 2022*). Bridging fibers fluxed faster than k-fibers in control and augmin-depleted cells (*Risteski et al., 2022*), supporting the model in which poleward flux is largely driven by sliding apart of antiparallel microtubules (*Brust-Mascher et al., 2009*; *Mitchison, 2005*; *Miyamoto et al., 2004*). We propose that augmin depletion results in slower flux of bridging fibers because the remaining bridging microtubules are likely nucleated at the poles, where microtubule depolymerization mechanisms might curb poleward flux speed (*Ganem et al., 2005*). In contrast, PRC1 depletion does not affect the flux (*Risteski et al., 2022*; *Steblyanko et al., 2020*) even though it reduces bridging fibers (*Kajtez et al., 2016*; *Polak et al., 2017*), possibly because the remaining bridging microtubules are generated away from the poles via augmin and can thus flux freely. In sum, augmin ensures proper architecture and dynamics of the metaphase spindle largely through the nucleation of bridging fibers, which link sister k-fibers and ensure their proper shape and function.

## Augmin-depleted spindles contain fewer overlap bundles, which have longer overlap regions and are located at the spindle periphery

Our finding that bridging fibers were more severely perturbed in the inner part of the spindle after augmin depletion prompted us to examine the spatial distribution of these midplane-crossing microtubules and their overlap regions throughout the spindle. We used protein regulator of cytokinesis 1 (PRC1) as a marker because it preferentially crosslinks overlap microtubules (*Li et al., 2018*; *Mollinari et al., 2002*), thus providing a specific label for bridging fibers (*Polak et al., 2017*). By taking a standard 'side view' of the spindle and rotating the 3D image stack of the spindle into an 'end-on' view, we were able to gain insight into the redistribution of bridging microtubules throughout the spindle cross-section in HeLa (*Kajtez et al., 2016*) and RPE1 (*Asthana et al., 2021*) cells stably expressing PRC1-GFP with and without MG-132 treatment (*Figure 4A–B*, *Figure 4—figure supplement 1A*). To compare their distribution to that of tubulin, we also rotated the 3D image stacks of the spindles in RPE1 cells stained with SiR-tubulin (*Figure 4A–B*, *Figure 4—figure supplement 1B*; *Novak et al., 2018*).

The signal intensity of PRC1-GFP bundles in RPE1 cells was reduced by 55 ± 4% following augmin depletion (*Figure 4C*). Consistently, the number of PRC1-labeled overlap bundles measured in an end-on view of spindles was almost halved; from 28±1–16±1 distinct bundles in control and HAUS6-depleted RPE1 cells, respectively (*Figure 4D*). Comparable trends were also observed in HeLa cells after depletion of HAUS6 or HAUS8 (*Figure 4—figure supplement 1C-G*).

The augmin-depleted cells showed a specific barrel-like distribution of the PRC1-GFP labeled bundles, with more overlap bundles being present around the perimeter of the spindle and fewer in the central part (*Figure 4A* 'end-on view' and 4D, *Figure 4—figure supplement 1H-I*). However, DNA was uniformly distributed throughout the spindle cross-section, both in augmin-depleted and control cells (*Figure 4A* 'end-on view'). In agreement with this result, kinetochores and tubulin signal were also found uniformly distributed over the spindle cross-section (*Figure 4A* 'end-on view' of RPE1 cells). This observation indicates that k-fibers are present and roughly uniformly distributed throughout the spindle cross-section and is in agreement with our finding that augmin primarily affects bridging fibers, while k-fibers are less perturbed (*Figure 3*).

To explore the role of the observed overlap repositioning in defining the overall spindle geometry, we measured spindle width, the diameter of the metaphase plate, spindle length, and overlap length in RPE1 and HeLa cells (see Materials and methods). Despite the spindles being wider in both

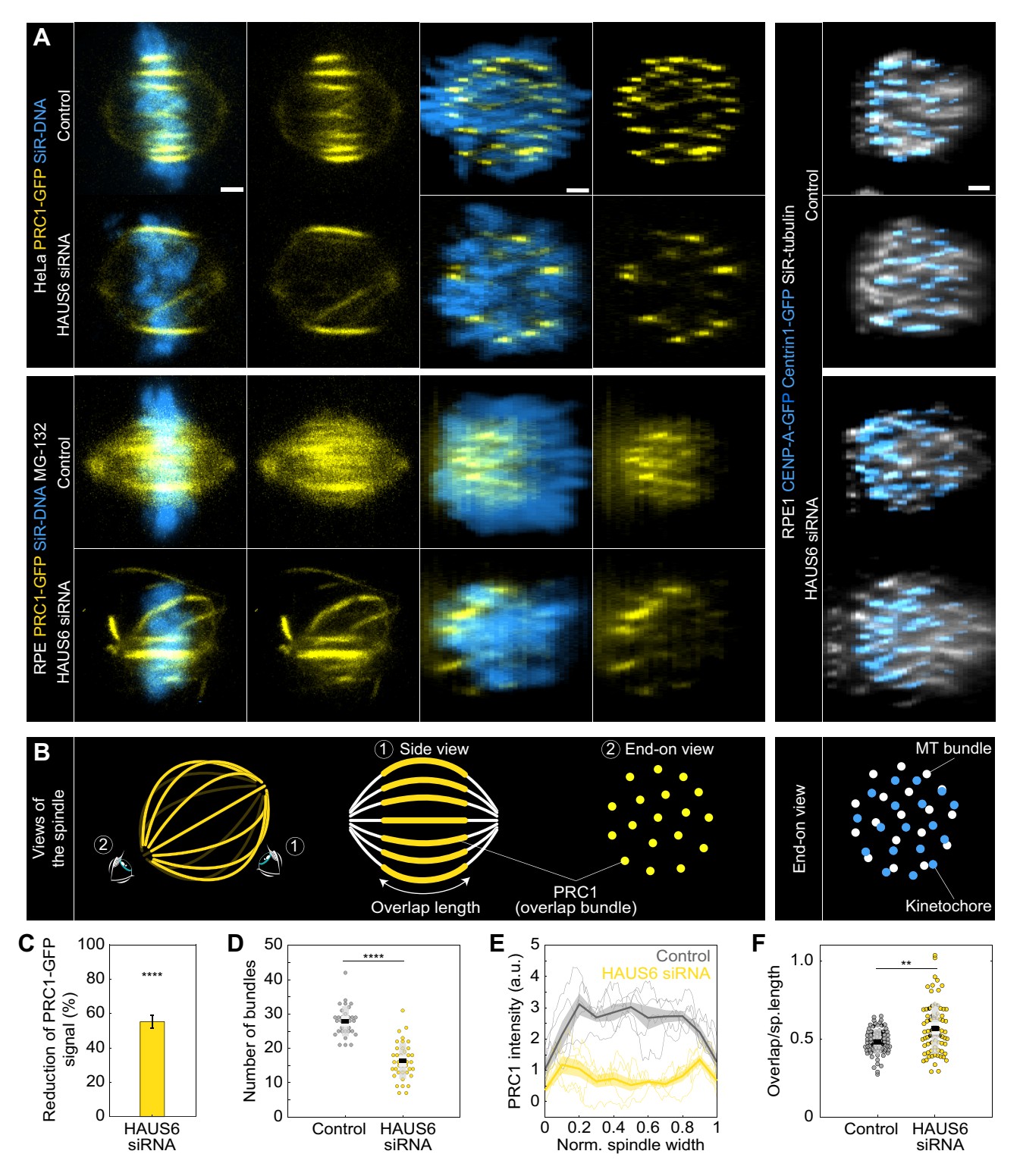

**Figure 4.** Augmin-depleted spindles have fewer bridging fibers, which have larger overlap length and are located at the spindle periphery. (**A**) The four columns on the left represent live images of metaphase spindles in untreated HeLa or MG-132-treated RPE1 cells stably expressing PRC1-GFP (yellow) and stained with SiR-DNA (blue) in control cells (top rows) and after HAUS6 depletion (bottom rows). 1st and 2nd column: side view of the spindle; 3rd and 4th column: end-on view of the same spindle, showing a barrel-like arrangement of PRC1-labeled bundles after augmin depletion. Images on the right show the end-on view of RPE1 cells stably expressing CENP-A-GFP and Centrin1-GFP (both in blue) and stained with SiR-tubulin (gray) in control

*Figure 4 continued on next page*

Figure 4 continued

cells (top) and after HAUS6 depletion (bottom). Side views are sum intensity projections of 5 central z-slices (Δz=0.5 µm) in HeLa cells and 10 central z-slices for RPE1 cells. End-on views are sum projections of 10 central z-slices (Δz=0.083 µm) for HeLa and 20 central z-slices for RPE1 cells. (**B**) Left: schematic representations of different views of the spindle. Eye signs mark the angle for the side view (1) and the end-on view (2). Side view was used to measure the length of overlap regions (yellow) and end-on view to determine the number of bundles (yellow dots). Right: schematic representation of the end-on view of RPE1 cells stably expressing CENP-A-GFP and Centrin1-GFP (blue dots) and stained with SiR-tubulin (gray dots). (**C**) The reduction of the PRC1 signal in RPE1 cells treated with MG-132 measured in sum intensity projection of 10 central z-slices following HAUS6 depletion. Values are shown as mean ± SEM. P-values were calculated using the absolute values of PRC1 signal intensity following HAUS6 depletion (N=39 cells), compared to the absolute values of PRC1 signal intensity in control cells (N=32 cells). (**D**) Univariate scatter plot of the number of bundles in RPE1 cells treated with MG-132 counted in the end-on view of the spindle in control cells (gray) and HAUS6-depleted cells (yellow). N=32 control cells and N=39 HAUS6-depleted cells. (**E**) The PRC1-GFP intensity profiles in RPE1 cells treated with MG-132 measured in the end-on view of the spindle in control cells (gray) and after HAUS6 depletion (yellow). The blue line in the inset marks the measured region (width: 2.5 µm). Mean (thick lines) and SEM (shaded areas). (**F**) Univariate scatter plot of overlap length divided by spindle length in RPE1 cells treated with MG-132 measured in the side view of the spindle in control cells (gray) and HAUS6-depleted cells (yellow). N=75 bundles in 32 control cells and N=74 bundles from 39 HAUS6-depleted cells. (**D and F**) Boxes represent standard deviation (dark gray), 95% confidence interval of the mean (light gray) and mean value (black). All results were obtained from three independent experiments. Statistical analysis (**C, D and F**) Mann–Whitney U test; p-value legend:<0.0001 (****), 0.0001–0.001 (***), 0.001–0.01 (**), 0.01–0.05 (*), ≥0.05 (ns). All images are adjusted for clarity so that all PRC1 bundles are visible in each cell (see Materials and methods). Scale bars, 2 µm.

The online version of this article includes the following figure supplement(s) for figure 4:

**Figure supplement 1.** Spindles without augmin are wider with fewer overlaps, which occupy a larger portion of the spindle length and accumulate at the spindle periphery.

cell lines, the diameter of the metaphase plate was not larger, as the spindles widened due to the long, curved bundles without kinetochores (*Figure 4—figure supplement 1J*). While the spindles in RPE1 cells shortened following augmin depletion, those in HeLa cells were longer (*Figure 4—figure supplement 1J*), consistent with previous observations on *Drosophila* S2 cells and *Xenopus* egg extracts (*Goshima et al., 2007*; *Petry et al., 2011*). This difference in spindle length might be due to the overlaps remaining the same length after augmin depletion in RPE1 cells, while being longer and thereby able to push the spindle poles further apart in HeLa cells (*Figure 4—figure supplement 1K*). When both spindle length and overlap length were taken into account, the relative length of overlaps with respect to spindle length increased in RPE1 cells from 48±1%–57 ± 2% following augmin depletion (*Figure 4F*), comparable to the increase in HeLa cells (*Figure 4—figure supplement 1L-M*). Altogether, these results suggest that augmin regulates both the width and length of metaphase spindles, while also restricting the portion of spindle length occupied by overlap microtubules.

Interestingly, the long curved bundles characteristic for augmin depletion (*Goshima et al., 2008*; *Uehara et al., 2016*; *Wu et al., 2008*; *Zhu et al., 2008*) exhibited PRC1 signal along most of their length, suggesting that they consist of antiparallel microtubules, even though contrary to bridging fibers, they form away from the k-fibers and kinetochores (corresponding to long, curved bundles in *Figure 1A* and *Figure 3—figure supplement 1*). These bundles likely arose either due to PRC1 cross-linking excessively long astral microtubules that were now able to reach the spindle midzone or due to PRC1 activity combined with the excess of free tubulin present as a consequence of less tubulin being incorporated in bridging and k-fibers. Altogether, the data suggest that there was an overall redistribution of PRC1 within the spindle from a large number of relatively short overlaps to a small number of relatively long overlaps. These results were further corroborated by PRC1-antibody staining in unlabeled HeLa cells, which also showed a reduced number of elongated PRC1 signals along the curved outer bundles after augmin depletion (*Figure 4—figure supplement 1F*). Thus, without augmin, the spindles are wider and contain fewer overlaps, which occupy a larger portion of spindle length and tend to accumulate at the spindle periphery.

## The interkinetochore distance decreases preferentially in the inner part of the spindle and at kinetochores with weaker bridging fibers after augmin depletion

The interkinetochore distance, which is a readout of interkinetochore tension (*Waters et al., 1996*), decreases after augmin depletion (*Uehara et al., 2009*; *Zhu et al., 2008*). Our measurements on RPE1 and HeLa cells also showed a reduced interkinetochore distance in augmin-depleted cells

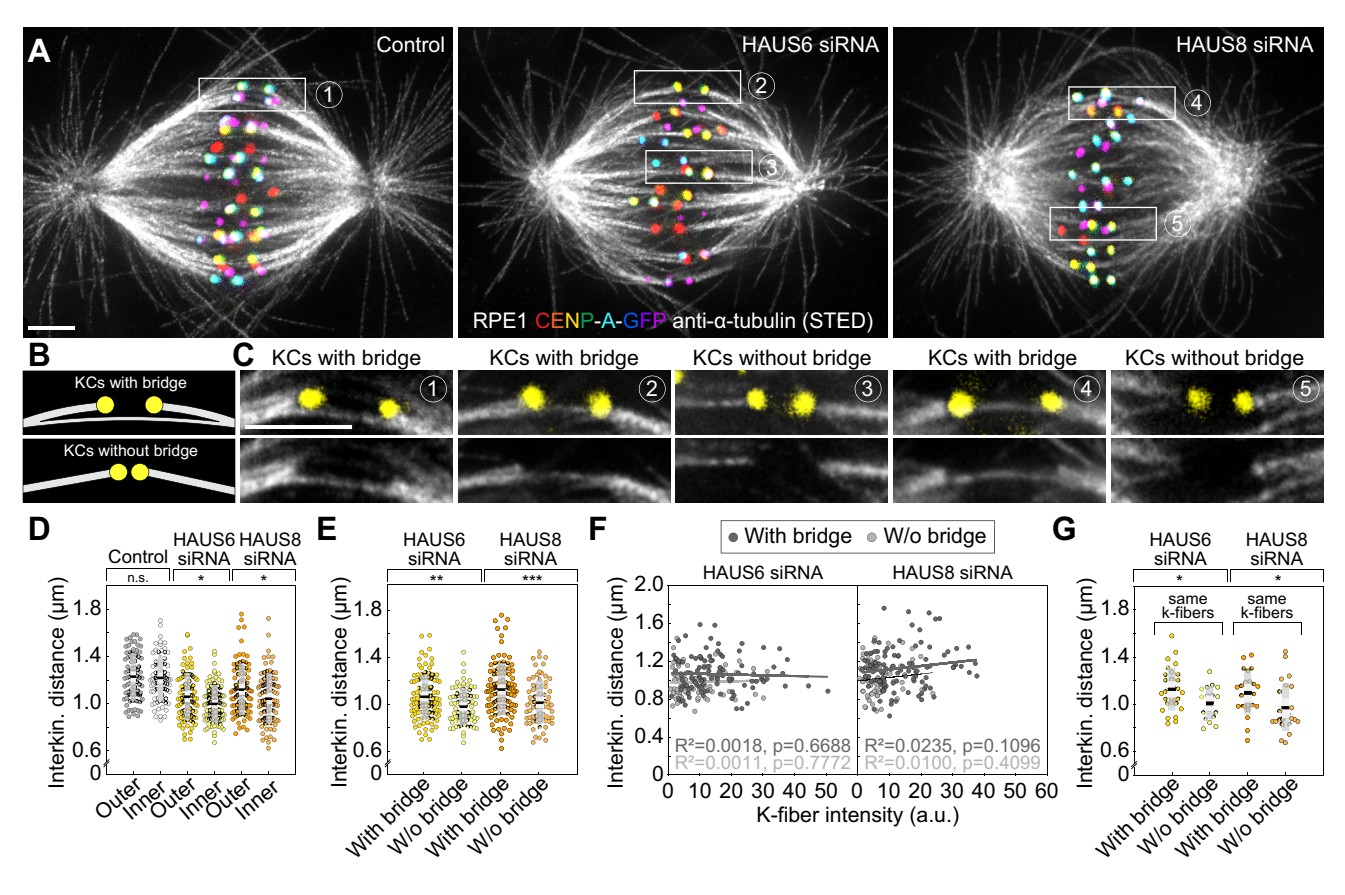

**Figure 5.** The reduction of the interkinetochore distance after augmin depletion is related to the impairment of bridging fibers. (**A**) STED superresolution images of microtubules immunostained for α-tubulin (gray) in control (left), HAUS6- (middle) and HAUS-8 depleted (right) RPE1 cells stably expressing CENP-A-GFP (rainbow, confocal). Images are maximum intensity projections and kinetochores are color-coded for depth from blue to red with the Spectrum LUT in ImageJ. (**B**) The schematic representation of a kinetochore pair (KCs) with (top) and without (bottom) bridging fiber (See Results). (**C**) Enlarged boxes show KCs with or without a bridging fiber in control (left), HAUS6- (middle), and HAUS8- (right) depleted RPE1 cells. Images represent single z-plane taken from spindles in (**A**) and smoothed with 0.75-mm-sigma Gaussian blur. Kinetochores are shown in yellow. (**D**) Univariate scatter plot of the interkinetochore distance in control (gray), HAUS6- (yellow), and HAUS8- (orange) depleted cells with kinetochore pairs divided based on their distance from the long (pole-to-pole) spindle axis (outer kinetochore pairs shown in darker colors and inner in lighter colors). N=30 cells in all three conditions; 78 and 80 outer and inner kinetochore pairs for control, respectively; 84 and 96 outer and inner kinetochore pairs for HAUS6 depletion, respectively; 88 and 92 outer and inner kinetochore pairs for HAUS8 depletion, respectively. (**E**) Univariate scatter plot of the interkinetochore distance in HAUS6- (yellow) and HAUS8- (orange) depleted cells. Kinetochore pairs are divided into two groups: with bridging fiber (darker colors) and without bridging fiber (lighter colors). N=30 cells in HAUS6/8-depleted cells; 106 pairs with and 74 kinetochore pairs without bridging fibers in cells following HAUS6 depletion, respectively; 110 and 70 kinetochore pairs with and without bridging fibers in cells following HAUS8 depletion, respectively. (**F**) The correlation of the interkinetochore distance and the k-fiber intensity for kinetochore pairs with (dark gray) and without (light gray) bridging fiber in HAUS6- (left) and HAUS8-depleted (right) cells. (**G**) Univariate scatter plot of the interkinetochore distance for HAUS6- (yellow) and HAUS8-depleted (orange) cells. Kinetochore pairs are divided into two groups: with bridging fiber (darker colors) and without bridging fiber (lighter colors), but in this case both groups have the same k-fiber intensity. N=27 kinetochore pairs with a bridging fiber and 18 kinetochore pairs without a bridging fiber in HAUS6-depleted cells, respectively; N=23 kinetochore pairs with and N=25 kinetochore pairs without a bridging fiber in HAUS8-depleted cells. (**D, E and G**) Boxes represent standard deviation (dark gray), 95% confidence interval of the mean (light gray) and mean value (black). All results were obtained from three independent experiments. Statistical analysis (**D, E and G**) t-test for samples that followed normal distribution or Mann–Whitney U test for samples that significantly departured from normality, determined using the Shapiro-Wilk test; (**F**) linear regression; p-value legend:<0.0001 (****), 0.0001–0.001 (***), 0.001–0.01 (**), 0.01–0.05 (*), ≥0.05 (ns). All images are adjusted for clarity based on the intensity of astral microtubules in each image (see Materials and methods). Scale bars, 2 µm.

The online version of this article includes the following figure supplement(s) for figure 5:

**Figure supplement 1.** Augmin-generated bridging fibers have a distinct role in the regulation of kinetochore tension.

(*Figure 5—figure supplement 1A*). This reduction of interkinetochore tension may be due to weaker k-fibers (*Uehara et al., 2009*; *Zhu et al., 2008*). However, we noticed that the interkinetochore distance was smaller in the inner part of the spindle in augmin-depleted cells (*Figure 5A–D*, *Figure 5—figure supplement 1B*), where bridging fibers were most severely impaired (*Figures 3H and 4A*). This was not the case in control cells, which showed no difference in interkinetochore distance between the inner and the outer part of the spindle (*Figure 5D*, *Figure 5—figure supplement 1B*). These findings motivated us to investigate a potential link between the lack of proper bridging fibers and the inter-kinetochore tension. We thus divided kinetochore pairs in STED images into two groups: (1) those with a bridging fiber (i.e. signal intensity of the bridging fiber above the background signal), and (2) those with undetectable signal intensities at the expected locations of bridging fibers, which we for simplicity refer to as kinetochore pairs without bridging fibers (*Figure 5B–C*). Remarkably, kinetochore pairs without bridging fibers had a significantly smaller interkinetochore distance than kinetochore pairs with bridging fibers (*Figure 5E*).

Although this result suggests a role of bridging fibers in regulating interkinetochore distance, this effect may be indirect and arise due to k-fibers, askinetochore pairs that lacked a bridging fiber typically had thinner k-fibers than those with a bridging fiber in augmin-depleted cells (*Figure 5—figure supplement 1C*). Hence, we used several approaches to separate the contribution of bridging and k-fibers to the interkinetochore tension. First, we found that although the interkinetochore distance correlated both with bridging and k-fiber intensity after augmin depletion, the correlation with bridging fiber intensity was stronger (*Figure 5—figure supplement 1D-E*). Such correlations were absent in control cells (*Figure 5—figure supplement 1D-E*). To explore a specific contribution of k-fibers to the interkinetochore tension, we divided the kinetochore pairs in augmin-depleted cells into two subsets, those with and without bridging fibers, and found that the interkinetochore distance did not correlate with k-fiber intensity within each group (*Figure 5F*), which argues against the k-fiber intensity as a sole determinant of interkinetochore tension. In agreement with this, when we selected two subsets of kinetochore pairs with either very strong or very weak k-fiber intensity but with comparable bridging fiber intensities (*Figure 5—figure supplement 1F-G*), we found no difference in the interkinetochore distance between these subsets (*Figure 5—figure supplement 1H*). Finally, to examine a specific contribution of bridging fibers, we identified two subsets of kinetochore pairs with similar k-fiber intensity values, one of which had bridging fibers and the other which did not (*Figure 5—figure supplement 1*). We found that the interkinetochore distance was larger in the subset with bridging fibers than without (*Figure 5G*), indicating a specific effect of bridging fibers on interkinetochore tension. Analysis of live-cell confocal images of RPE1 cells yielded similar results (*Figure 5—figure supplement 1J-M*). Based on these data, we conclude that augmin has a significant role in regulating interkinetochore tension through the nucleation of bridging microtubules.

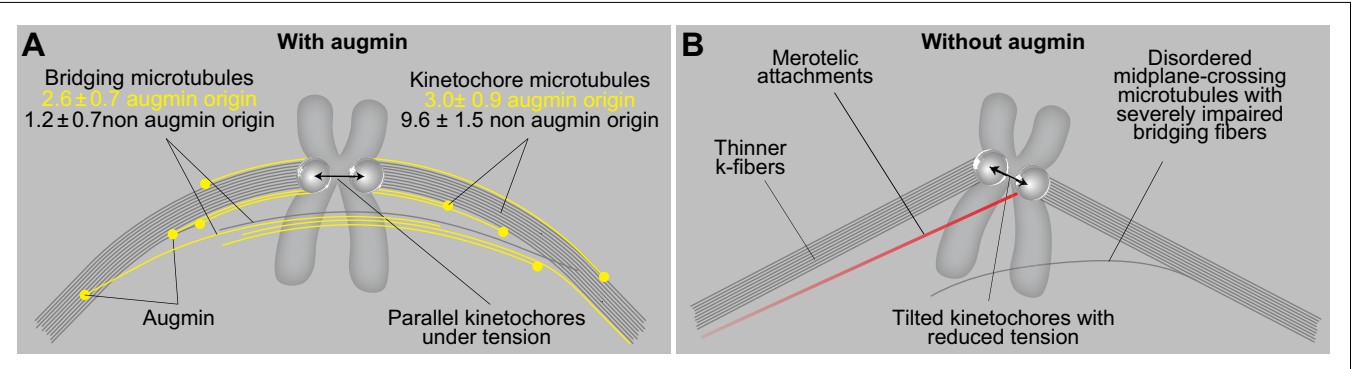

**Figure 6.** A model of augmin-dependent nucleation of bridging microtubules with their contribution to mitotic fidelity. (**A**) Bridging microtubules are to a large extent formed by augmin-dependent nucleation. They ensure the alignment of sister kinetochores parallel to the spindle axis and the interkinetochore tension. Augmin-nucleated microtubules (yellow) and microtubules nucleated through other pathways (gray) in bridging and k-fibers are shown together with the number of microtubules in each group, as estimated from HAUS6 depletion experiments (See Results). (**B**) Impaired structure of bridging fibers upon augmin depletion leads to weaker interkinetochore tension and increased tilt of the kinetochores, which puts kinetochores at risk of interacting with additional microtubules (red), resulting in merotelic attachments.

## Discussion

In this paper, we propose a model of the metaphase spindle in which the bridging fiber, which laterally connects sister k-fibers, forms by augmin-based nucleation of microtubules along the existing microtubules. The newly nucleated microtubules in the bridging fiber create an antiparallel overlap in which the microtubules slide apart, generating a pushing force that the bridging fiber exerts on its k-fibers. In doing so, the bridging fiber works together with k-fibers to produce tension and maintain the appropriate orientation of sister kinetochores parallel to the spindle axis, thereby preventing merotelic attachments and ensuring mitotic fidelity (*Figure 6*).

Our work shows that the depletion of the augmin complex by silencing the HAUS6 or HAUS8 subunits causes severe thinning of bridging fibers in metaphase spindles, combined with the appearance of wavy and disordered midplane-crossing microtubules. Thus, we conclude that the predominant nucleation of bridging microtubules by the augmin complex enables their directional bundling along the sister k-fibers, in agreement with previous observations on directionality of microtubule growth (*David et al., 2019*; *Kamasaki et al., 2013*). K-fibers were also thinner, though to a lesser extent, indicating that they are largely nucleated in an augmin-independent manner, at the centrosome or kinetochores and chromosomes. This is in agreement with previous electron microscopy studies of mammalian spindles, where k-fiber microtubules were observed to typically reach the centrosome, while sometimes also extending from the kinetochores with their minus ends within the spindle (*Kiewisz et al., 2022*; *McDonald et al., 1992*; *Sikirzhytski et al., 2018*), the latter likely representing a combination of microtubules nucleated either at the kinetochores or by the augmin complex. In contrast, most midplane-crossing microtubules, which likely correspond to bridging fibers, start at different points along the k-fiber (*Mastronarde et al., 1993*). Electron tomography of spindles in RPE1 cells confirmed this result by showing that microtubule minus ends are found along the k-fiber, less than 50 nm from the k-fiber wall and at a distance 2–4 μm from the pole (*O'Toole et al., 2020*). As we found that the same number of microtubules in bridging and k-fibers was nucleated by augmin, we propose that the existing microtubules orient the growth of augmin-nucleated microtubules (*David et al., 2019*; *Kamasaki et al., 2013*), which later become kinetochore microtubules if their plus end binds to the kinetochore or bridging microtubules if they grow past the kinetochore and interact with the bridging or kinetochore microtubules on the other side. However, as there are less microtubules in the bridging fiber to begin with, augmin-mediated nucleation becomes the predominant pathway of their nucleation.

Previous work showed that augmin depletion results in a decrease of interkinetochore distance (*Uehara et al., 2009*; *Zhu et al., 2008*), a readout of tension, but it remained unclear if this is due to impaired k-fibers or perturbation of other microtubules. Treatment of cells with a microtubule-destabilizing agent that results in thinner k-fibers causes a reduction of the interkinetochore tension (*Dudka et al., 2018*), supporting the former possibility. However, a similar effect on the interkinetochore tension was observed upon perturbation of the bridging fiber by removing the microtubule crosslinker PRC1 (*Jagrić et al., 2021*; *Kajtez et al., 2016*; *Polak et al., 2017*), in agreement with the latter possibility. When looking at a subset of kinetochore pairs that had a bridging fiber and those that did not, we found that the tension was more compromised in the latter group. Notably, tension was independent of the k-fiber thickness within each group and depended on the presence or the absence of bridging fibers even when the effect of k-fibers was excluded. Although our experiments cannot directly discern the exact contribution of the bridging and k-fiber impairment in the decrease of tension on kinetochores, they reveal that augmin-generated bridging microtubules have a significant and specific role in the maintenance of interkinetochore tension.

Considering the importance of the interkinetochore tension for the accuracy of cell division (*Lampson and Grishchuk, 2017*), the maintenance of tension by the bridging fiber might represent an important mechanism for silencing of the spindle assembly checkpoint (*Musacchio and Salmon, 2007*; *Nicklas et al., 1995*), supported by the fact augmin-depleted cells with characteristically compromised bridging fibers have extensive mitotic delays (*Wu et al., 2008*). However, recent work suggests that the reduction of interkinetochore tension caused by k-fiber thinning does not necessarily lead to checkpoint activation, but instead may sometimes result in reduced anaphase A speed caused by low microtubule occupancy, followed by a subsequent increase in lagging chromosomes (*Dudka et al., 2018*). As proteins involved in bridging fiber nucleation and crosslinking, including HAUS6, HAUS8, and PRC1 have all been linked to various types of cancer (ICGC/TCGA, 2020, retrieved by using

cBioPortal *Cerami et al., 2012*; *Gao et al., 2013*), it is plausible that impairment of bridging fibers also leads to such increase in lagging chromosomes and contributes to aneuploidy in cancers. Indeed, once the checkpoint was weakened, lagging chromosomes that appeared following augmin depletion had a reduced interkinetochore distance, which was consistent with previous findings on lazy chromosomes (*Sen et al., 2021*). However, contrary to previous findings in cells with intact bridging fibers, in which there was no connection between lagging chromosomes and tilt of their kinetochores (*Sen et al., 2021*), we found that kinetochores of lagging chromosomes in augmin-depleted cells predominantly form a large tilt with respect to the spindle axis. This suggested that the lack of bridging fibers represents a specific mechanism of predisposing the kinetochore to merotelic attachments by tilting the kinetochores and exposing their surface to the microtubules from the opposite pole. The tilted kinetochore pair is more likely to encounter additional microtubules also because midplane-crossing microtubules that form following augmin depletion no longer exhibit directional growth, but are instead wavy and extend in various directions. Once a merotelic attachment forms, it may further amplify the existing tilt due to pulling by additional microtubules in a skewed direction. Moreover, erroneous attachments might be less likely to undergo error correction in augmin-depleted cells, as bridging fibers may serve as tracks that guide Aurora B within the Chromosome Passenger Complex towards the kinetochores to correct the errors (*Matković et al., 2022 Preprint*). In addition to predisposing the kinetochore to merotelic attachments through impaired bridging fibers, thinning of k-fibers upon augmin depletion might ultimately be responsible for the inability to resolve merotelic attachments, as strong k-fibers are necessary to ensure proper segregation of kinetochores during anaphase (*Dudka et al., 2018*). Altogether, our results reveal that augmin is the leading nucleator of bridging fibers and suggest a delicate interplay of bridging and k-fibers in the maintenance of spindle organization, kinetochore tension and mitotic fidelity.

## Materials and methods

### Cell lines

Experiments were carried out using unlabeled human HeLa-TDS cells from the High-Throughput Technology Development Studio (MPI-CBG, Dresden); human HeLa-Kyoto BAC cells stably expressing PRC1-GFP (*Poser et al., 2008*), courtesy of Ina Poser and Tony Hyman (MPI-CBG, Dresden, Germany); human HeLa cells stably expressing CENP-A-GFP and Centrin1-GFP (*Gasic et al., 2015*), which were a gift from Emanuele Roscioli and Andrew McAinsh (University of Warwick, Coventry, UK); human HeLa-TDS cells stably expressing GFP-α-tubulin as described in our previous work *Kajtez et al., 2016*; human hTERT-RPE1 (hTERT immortalized retinal pigment epithelium) cells stably expressing CENP-A-GFP and human hTERT-RPE1 (hTERT immortalized retinal pigment epithelium) cells stably expressing both CENP-A-GFP and Centrin1-GFP (*Magidson et al., 2011*), a courtesy of Alexey Khodjakov (Wadsworth Center, New York State Department of Health, Albany, NY, USA); human RPE1 CRISPR-Cas9 cells stably expressing PRC1-GFP (*Asthana et al., 2021*), a gift from Thomas Surrey (Centre for Genomic Regulation, Barcelona, Spain); and human U2OS cells stably expressing CENP-A-GFP, mCherry-α-tubulin and PA-GFP-α-tubulin (*Barisic et al., 2014*), a gift from Marin Barišić (Danish Cancer Society Research Center, Copenhagen, Denmark).

### Cell culture

All cell lines were cultured in flasks in Dulbecco's Modified Eagle's Medium with 1 g/L D-glucose, pyruvate and L-glutamine (DMEM, Lonza, Basel, Switzerland), supplemented with 10% (vol/vol) heat-inactivated Fetal Bovine Serum (FBS, Sigma Aldrich, St. Louis, MO, USA) and penicillin (100 IU/mL)/streptomycin (100 mg/mL) solution (Lonza, Basel, Switzerland). For the selection of U2OS CENP-A-GFP mCherry-α-tubulin PA-GFP-α-tubulin, HeLa PRC1-GFP, HeLa CENP-A-GFP Centrin1-GFP and Hela GFP-α-tubulin cell lines, 50 µg/ml geneticin was added to the medium (Life Technologies, Waltham, MA, USA).

All cells were kept at 37 °C and 5% $CO_2$ in a Galaxy 170 R humidified incubator (Eppendorf, Hamburg, Germany). They have also been regularly tested for mycoplasma contamination by examining the samples for extracellular DNA staining with SiR-DNA (100 nM, Spirochrome, Stein am Rhein, Switzerland) and Hoechst 33342 dye (1 drop/2 ml of NucBlue Live ReadyProbes Reagent, Thermo Fisher Scientific, Waltham, MA, USA) and have been confirmed to be mycoplasma free.

## Sample preparation and RNAi transfection

At 80% confluence, DMEM medium was removed from the flask and cells were washed with 5 ml of PBS. Then, 1 ml 1% trypsin/EDTA (Biochrom AG, Berlin, Germany) was added to the flask and cells were incubated at 37 °C and 5% $CO_2$ in a humidified incubator for 5 min. After the incubation, trypsin was blocked by adding 2 ml of DMEM medium. For RNAi experiments, the cells were seeded to reach 60% confluence the next day and cultured on 35 mm uncoated dishes with 0.16–0.19 mm (1.5 coverglass) glass thickness (MatTek Corporation, Ashland, MA, USA) in 2 mL DMEM medium with previously described supplements. After one day of growth, cells were transfected with either targeting or non-targeting siRNA constructs which were diluted in OPTI-MEM medium (Life Technologies, Waltham, MA, US) to a final concentration of 20 nM for HAUS6 and HAUS8 in medium with cells. HAUS6 and HAUS8 siRNA transfections were performed 48 hr prior to imaging and cotransfection with 100 nM Mad2 siRNA was additionally performed 24 hr prior to imaging using Lipofectamine RNAiMAX Reagent (Life Technologies, Waltham, MA, US) according to the instructions provided by the manufacturer. After four hours of treatment, the medium was changed to the previously described DMEM medium. The constructs used were human HAUS6 siRNA (L-018372-01-0005, Dharmacon, Lafayette, CO, USA), human HAUS8 siRNA (L-031247-01-0005, Dharmacon, Lafayette, CO, USA), human Mad2 siRNA (L-003271-00-0010, Dharmacon, Lafayette, CO, USA) and control siRNA (D-001810-10-05, Dharmacon, Lafayette, CO, USA). Mad2 was chosen instead of MPS1 inhibitors as it induces less segregation errors per cell in control cells, as well as to avoid a biased approach in which the inhibitor would likely be added at a different time point during an hours-long error correction process that occurs in augmin-depleted cells.

When microtubules were visualized during live cell-imaging in cell lines without stable expression of tubulin, staining was performed to a final concentration of 100 nM with a far-red silicon rhodamine (SiR)-tubulin-670 dye (Spirochrome, Stein am Rhein, Switzerland), 45 min to 2 hr prior to imaging. As for DNA, either SiR-DNA or SPY-555-DNA (Spirochrome, Stein am Rhein, Switzerland) were used to a final concentration of 100 nM, 20 min to 2 hr prior to imaging. In order to avoid dye efflux, a broad-spectrum efflux pump inhibitor verapamil (Spirochrome, Stein am Rhein, Switzerland) was added at a final concentration of 0.5 µM to RPE1 cells along with tubulin and/or DNA dyes. As an additional control for metaphase cells, MG-132 (Sigma Aldrich, M7449-1ML, MO, USA) was added to the cells used for the measurement of spindle size for 30 min prior to imaging at a final concentration of 20 µM.

## Immunofluorescence

For confocal imaging, HeLa cells stably expressing PRC1-GFP were grown on glass-bottom dishes (14 mm, No. 1.5, MatTek Corporation) and fixed by 2 ml of ice-cold methanol for 1 min at –20 °C. Following fixation, cells were washed three times for 5 min with 1 ml of PBS and permeabilized with 0.5% Triton-X-100 in water for 15 min at a room temperature. This step was repeated twice when tubulin staining was performed. To block unspecific binding, cells were incubated in 1 ml of blocking buffer (1% normal goat serum (NGS)) for 1 hr at 4 °C. Cells were then washed three times for 5 min with 1 ml of PBS and incubated with 250 µl of primary antibody solution overnight at 4 °C. The primary antibodies used were as follows: rabbit polyclonal PRC1 (diluted 1:100, sc-8356, Santa Cruz Biotechnology, RRID:AB_2169665), rabbit polyclonal HAUS6 (diluted 1:250, ab-150806, Abcam), rabbit polyclonal HAUS8 (diluted 1:100, PA5-21331, Invitrogen, RRID:AB_11153508) and rat monoclonal tubulin (diluted 1:100, MA1-80017, Invitrogen, RRID:AB_2210201). After the incubation with a primary antibody, cells were washed 3 times for 5 min with 1 ml of PBS and then incubated with 250 µl of secondary antibody for 45 min at a room temperature. Alexa Fluor 488 and 594 (Abcam, ab150073, ab150076, ab150156) were used as secondary antibodies at a 1:1000 dilution for PRC1 staining, 1:500 dilution for HAUS6 and 1:250 for HAUS8 and tubulin staining. DAPI (1 µg/mL) was used for chromosome visualization.

For STED imaging, RPE1 cells stably expressing CENP-A-GFP were grown on glass-bottom dishes (14 mm, No. 1.5, MatTek Corporation), cell medium was removed, and cytoskeleton extraction buffer was added for 20 s to remove the components of the cytoplasm. Following extraction, cells were fixed in 3% paraformaldehyde and 0.1% glutaraldehyde solution for 10 min. To reduce the background fluorescence, quenching (100 mM glycine in PBS) and reduction (0.1% sodium borohydride in PBS) solution were added for 7 and 10 min, respectively. To prevent non-specific binding, cells were incubated in blocking/permeabilization buffer (2% normal goat serum and 0.5% Triton-X-100 in PBS) for

2 hr at 4 °C. Microtubules were then stained using a rat anti-tubulin primary antibody solution (diluted 1:500 in blocking/permeabilization buffer, MA1-80017, Invitrogen) with a 4 °C overnight incubation. The next day, cells were washed with PBS three times for 5 min. After washing, a secondary antibody Alexa Fluor 594 (dilution 1:1000, Abcam) was added and incubated for 1 hr at room temperature. Additionally, DAPI (1 µg/mL) was added and incubated for 15 min at room temperature to visualize chromosomes. Cells exposed to cold treatment were incubated with cold media on ice for 13 min prior to extraction of cytoplasmic components.

## Immunoblotting

RPE1 cells stably expressing CENP-A-GFP and Centrin1-GFP were grown on six-well plates (Greiner Bio-one) and subjected to HAUS6 or HAUS8 siRNA treatment as indicated before. Following transfection, cells were lysed in RIPA buffer (R0287, Sigma) containing 1 x protease inhibitor (5892970001, Roche), 1 x phosphatase inhibitor (4906837001, Roche) and 1 mM PMSF by two cycles of freezing and thawing in liquid nitrogen. Protein extracts were mixed with 2 x Laemlli sample buffer (S3401, Sigma) and heated at 95 °C for 10 min prior to SDS-PAGE. After protein transfer onto the nitrocellulose membrane (IB23002, Invitrogen) and blocking with blocking solution (5% bovine serum albumin and 0,1% Tween 20 in PBS) for 1 hr, membranes were incubated overnight at 4 °C with primary antibodies diluted in blocking solution. The primary antibodies used were as follows: rabbit polyclonal HAUS6 (diluted 1:1000, ab-150806, Abcam), rabbit polyclonal GAPDH (diluted 1:1000, ab9485, Abcam, RRID:AB_307275). Rabbit polyclonal HAUS8 antibody (diluted 1:1000, PA5-21331, Invitrogen, RRID:AB_11153508, and NBP2-42849, Novus Biologicals, RRID:AB_2665500) resulted in no detectable bands under these conditions. Membranes were washed with 0.1% Tween 20 in PBS, incubated for 1 hr with anti-rabbit HRP-conjugated secondary antibodies (dilution 1:10,000, ab6721, RRID:AB_955447) and visualized on the C-DiGit blot scanner (LI-COR, Bad Homburg, Germany) with WesternSure PREMIUM Chemiluminescent Substrate (926–95000, LI-COR).

## Microscopy

STED microscopy of fixed cells and live-cell imaging of anaphase were performed using an Expert Line easy3D STED microscope system (Abberior Instruments, Göttingen, Germany) with the 100 x/1.4NA UPLSAPO100x oil objective (Olympus, Tokio, Japan) and an avalanche photodiode (APD) detector. The 488 nm line was used for excitation in both cases, with the addition of the 561 nm line for excitation and the 775 nm laser line for depletion during STED superresolution imaging. Images were acquired using the Imspector software. The xy pixel size for fixed cells was 20 nm and 6 focal planes were acquired with 300 nm distance between planes were acquired. For confocal live-cell imaging of anaphase, the xy pixel size was 80 nm and 16 focal were acquired, with 1 µm distance between the planes and 30 s time intervals between different frames.

Other confocal images and videos were acquired using a previously described microscope setup (*Buđa et al., 2017*), consisting of a Bruker Opterra Multipoint Scanning Confocal Microscope (Bruker Nano Surfaces, Middleton, WI, USA), mounted on a Nikon Ti-E inverted microscope with a Nikon CFI Plan Apo VC 100 x/1.4 numerical aperture oil objective (Nikon, Tokyo, Japan). During live-cell imaging, cells were kept at 37 °C and 5% $CO_2$ in Okolab Cage Incubator (Okolab, Pozzuoli, NA, Italy). To excite Hoechst, GFP, mCherry or SiR fluorescence, a 405 nm, 488 nm, 561 nm or 647 nm laser lines were used, respectively. Opterra Dichroic and Barrier Filter Set 405/488/561/640 enabled the separation of excitation light from the emitted fluorescence. Images were acquired using Evolve 512 Delta Electron Multiplying Charge Coupled Device (EMCCD) Camera (Photometrics, Tuscon, AZ, USA), with camera readout mode of 20 MHz. The xy pixel size was 83 nm. In all experiments where the whole spindle stack was imaged, z-stacks were acquired with unidirectional xyz scan mode at 37 focal planes and 0.5 µm distance between the planes. Photoactivation was performed perpendicular to pole-to-pole axis of metaphase spindles using a 405 nm laser diode (Coherent, Santa Clara, CA, USA) and a line pattern of 12 equally distributed points, with each point representing one laser hit. The interval between the points was 0.05ms and photoactivation area was set to 0.5 µm for each point. The interval between successive frames was set to 10 s and one central z-plane was imaged.

## Image processing and data analysis

In cells where HAUS6 or HAUS8 were silenced only bipolar metaphase spindles were imaged and analyzed, even though multipolar spindles were observed as reported previously (*Lawo et al., 2009*). All images were analyzed in Fiji/ImageJ (National Institutes of Health, Bethesda, MD, USA, RRID:SCR_003070). Raw images were used for quantification. In representative immunofluorescence images of augmin depletion (*Figure 2—source data 1*), all signals were adjusted equally in control and treated cells. However, due to severe reduction of bridging fibers in augmin-depleted spindles, the contrast in images for representation on figures was not always equally adjusted, as this led to important spindle structures being either highly oversaturated or barely visible. It was instead adjusted so that astral microtubules are similarly visible in control and augmin-depleted spindles in STED microscopy, or that all present bridging fibers are visible in confocal microscopy. These adjustments did not result in any structures being omitted or otherwise modified in a way that could lead to misrepresentation. MatLab (MathWorks, Natick, MA, USA, RRID:SCR_001622) was used for calculations and plotting and Rstudio (R Foundation for Statistical Computing, Vienna, Austria, RRID:SCR_000432) to transform the cells into an end-on view. Figures were assembled in Adobe Illustrator CS5 (Adobe Systems, Mountain View, CA, USA, RRID:SCR_010279).

## Measuring the bridging fiber intensity

For the analysis based on small square regions which was performed on both STED and confocal images, tubulin intensity in bridging and k-fibers was measured using either a 25x25 or 5x5 pixel Square tool (ImageJ) on STED and confocal images, respectively. To measure bridging fibers, the square was positioned on the fiber spanning the area between two kinetochores. K-fibers were measured right next to the kinetochores. The average intensity of the two sister k-fibers was used for further analyses. The background was measured using the same tool at several empty areas within the spindle and its average was subtracted from bridging ($I_b = I_{b+bcg} - I_{bcg}$) and k-fiber intensities ($I_k = I_{k+bcg} - I_{bcg}$). 10 randomly positioned empty areas were taken into account while calculating background for STED imaging, whereas 2 empty areas just above and below bridging fiber were taken into account while calculating background for confocal imaging due to different nature of the acquired images. All measurements were performed on randomly selected bundles in single z-planes, after determining that no other microtubules were crossing the area of measurement. In STED images, the intensity of astral microtubules was additionally measured using the 25x25 pixel Square tool (ImageJ). Their background was measured using the same tool in the empty area between the two astral microtubules and it was subtracted from astral microtubule intensity ($I_a=I_{a+bcg}-I_{bcg}$).

On top of that, tubulin intensities of the bridging fiber and k-fiber region in confocal images were measured in a single z plane using the Segmented Line tool by drawing a 5 pixel line (ImageJ) along the contour of k-fibers and the corresponding bridging fiber. Background was measured in the same z plane by drawing the 5 pixel thick line along the length of the metaphase midzone, as this represents the background signal caused by the presence of neighboring microtubules. The minimum value of the midzone intensity profile was subtracted from the intensity profiles of bundle contours. The minimum value of the mean tubulin intensity profile was set as a distance of zero µm and was selected as the center of the bridging fiber. The final intensity of a bridging fiber ($I_b$) was calculated as the mean value of intensities in the area 500 nm around the center of the bridging fiber. The final intensity of a k-fiber region ($I_{bk}$), which also includes the bridging fiber, was calculated as an average of two mean values of intensities in the area 500 nm around the distance of 1.5 µm away from the center of the bridging fiber. The intensity value of k-fibers alone ($I_k$) was then calculated as $I_k = I_{bk} - I_b$.

## Comparison of SiR- and GFP-tubulin signal

For the comparison of SiR and GFP-tubulin signal in HeLa cells stably expressing GFP-α-tubulin, the previously described analysis was used. Tubulin intensity of the bridging fiber and k-fiber region was measured in a single z plane using the Segmented Line tool in ImageJ by drawing the 5 pixel line along the contour of k-fibers and the corresponding bridging fiber. Due to high noise, only outermost bundles were taken into analysis and the reduction was not calculated. The background was measured in the same z plane by drawing the 5x5 pixel square in the cytoplasm and was subtracted from the bundle intensity profiles. The minimum value of the intensity profile was set as a distance of zero µm and was selected as the center of the bridging fiber.

### The interkinetochore distance

To calculate the interkinetochore distance, two points were put on the centers of signal in each kinetochore pair using a Point tool in ImageJ. Additionally, two points were put on spindle poles and two points on the upper and lower edge of the metaphase plate. Interkinetochore distance and distance from spindle axes were calculated using a home-written MatLab script. Two-dimensional analysis was applied when all tracked kinetochore pairs resided within 2 µm around the central z-plane. Three-dimensional analysis was used when all kinetochore pairs in the spindle were taken into account, regardless of their position. In 3D analysis, the exact distance from the long spindle axis (c) was calculated using the Pythagorean theorem, where a=projected distance of a middle point between two sister kinetochores in a particular z-plane from the long spindle axis, b=distance between the central z-plane and the z-plane of the kinetochore pair x 0.81 (correction factor for the oil objective only, see Transformation of spindles into an end-on view). Kinetochore pairs were defined as those in the inner or the outer part of the spindle if their distance to the pole-to-pole axis was smaller or larger than the average distance of all tracked kinetochore pairs to the pole-to-pole axis, respectively. Additionally, kinetochores with visible bridging fibers and kinetochores with no visible bridging fibers were separately tracked and analyzed in confocal images of RPE1 cells stably expressing CENP-A-GFP and Centrin1-GFP, where the presence of bridging fibers was determined by measuring intensity profiles of the tubulin signal between two kinetochores.

### Transformation of spindles into an end-on view

Z-stacks of the mitotic spindles that were positioned horizontally were transformed into the end-on view using a home-written R script, based on the original version (*Novak et al., 2018*). Prior to transformation, a single-channel grayscale z-stack was rotated using ImageJ to make the long axis of the spindle parallel to the x-axis. Signal intensity at each pixel was used to obtain an end-on view of the spindles by applying the following transformation: I' (i · pixel size, j · pixel size, k · z-distance)=I (k · z-distance, I · pixel size, j · pixel size). A correction factor of 0.81 was used for the z-distance to correct for aberrations caused by the different refractive index mismatch of aqueous samples and immersion oil (*Novak et al., 2018*).

### Number of bundles

The number of bundles in HeLa and RPE1 PRC1-GFP cells was determined in an end-on view of the spindle by using sum intensity projections of 10 central z-planes covering 0.83 µm along the long spindle axis.

### Spindle length, width, and metaphase plate diameter

Spindle length, width and metaphase plate diameter were measured on maximum intensity projections of the side view z-stack of spindles. Spindle length was determined as a distance between the two poles. The position of the poles was determined as either the center of Centrin1 signal or the outermost points of the tubulin or PRC1 signal at the spindle pole. Spindle width was measured as the distance between two lines parallel to the long axis of the spindle and encompassing the outermost PRC1- or tubulin-labeled bundles. Additionally, in RPE1 cells stably expressing CENP-A-GFP and Centrin1-GFP the metaphase plate diameter was measured as the distance between the outermost kinetochore pairs, whereas in HeLa PRC1-GFP it was measured as the distance between the outermost chromosome ends.

### Overlap length

Overlap length was measured on sum intensity projections of 2–4 z-planes covering the entire bundle of interest, using ImageJ Segmented line tool by drawing a pole to pole line along the contour of PRC1-GFP and acquiring an intensity profile. The overlap length was defined as the length of the base of the PRC1-GFP intensity peak (*Polak et al., 2017*).

### PRC1 intensity

PRC1 intensity was measured in sum intensity projections of 10 central z-planes of the spindle. Total PRC1 signal in the cell was marked by using Polygon selection tool (ImageJ) and 5x5 Square tool was used to determine the background in the cytoplasm. The final intensity values were obtained using the

following formula: PRC1 intensity = Integrated Density of the spindle – (Area of selected cell x mean fluorescence of background). Intensity profiles of PRC1-GFP were measured on the sum intensity projections of 10 central z-planes in an end-on view of the spindle by drawing a 50 pixel wide Straight line tool across the diameter of the spindle.

### Spindle shape

The shape of spindles was determined in ImageJ using a Point tool. Ten points were distributed throughout the bundle, with the first and last point positioned at the spindle poles. The images were rotated to make the long spindle axis parallel to the x axis. In control cells, only the outermost bundle was tracked. In HAUS6 siRNA-treated cells, three different groups of outermost bundles were tracked: bundles with visible bridging fibers, bundles with no visible bridging fibers and curved bundles extending far from the metaphase plate. Shape and curvature were calculated using a homewritten MatLab script by fitting a circle to the tracked points along the bundle. Contour lengths of the bundles were measured by calculating the cumulative distance between the first and the last point of the tracked bundle.

### Poleward flux rate

For measuring the poleward flux rate, 10-pixel wide line was drawn from pole to pole along the bundle with photoactivation signal that lasted at least 5 time frames (40 s), using the Segmented Line tool in ImageJ. The position of photoactivated mark in each time frame was determined as the distance between the peaks of intensity profiles in photoactivation and SiR-tubulin channels for photoactivation mark and closer spindle pole, respectively. The analysis was performed on images processed with Gaussian Blur filter with Sigma set to 2 to improve the definition of the intensity profile peaks.

### Tracking and classification of segregation errors

Three types of segregation errors were analyzed in Mad2-depleted and Mad2/HAUS6-codepleted anaphase spindles: misaligned kinetochores, lagging kinetochores and other errors. Misaligned kinetochores were defined as those in which both kinetochores of the pair were situated outside the metaphase plate 30 s before anaphase onset. Lagging kinetochores were defined as those in which the CENP-A signal was visibly stretched and the kinetochore was situated in the central part of the spindle, outside the kinetochore mass that was moving towards the pole during anaphase. Finally, segregation errors classified as others included various kinetochores, precise classification of which was impossible using only CENP-A signal. These included kinetochores situated outside the moving kinetochore mass without stretched CENP-A signal, kinetochore pairs that remained completely unseparated for the whole duration of anaphase, as well as kinetochore pairs in which both kinetochores remained nonstretched and in the central part of the spindle despite the initial separation. Segregation errors were further divided based on their distance to the pole-to-pole axis into those in the inner and the outer part of the spindle. The distance of the kinetochore pair from the pole-to-pole axis was determined in 3D using a home written Matlab script, and the obtained value was then normalized to spindle half-width determined using maximum intensity projections in ImageJ. All kinetochores situated in the area less than 0.5 from the spindle axis 30 s before anaphase onset were defined as those in the inner part of the spindle and all kinetochore pairs in the area equal to or above the value of 0.5 were defined as kinetochores in the outer part of the spindle. All kinetochore pairs were manually tracked in time from just before anaphase onset until entering the daugther cell by using the Point tool in ImageJ. In images acquired using STED microscopy, merotelic attachments were defined as those in which one kinetochore forms attachments with microtubules from the opposite side of the spindle, with no visible microtubule signal just below or above the kinetochore.

### Anaphase A and B speed

To measure anaphase A speed, the coordinates of kinetochores and poles were tracked in time using the Point tool in ImageJ. The speed was calculated from the time point when the distance between the kinetochore and its closer pole started gradually decreasing. The slope of a line equation obtained from the linear fitting of distances over time was determined for every kinetochore as the anaphase A speed. Anaphase B speed was calculated in a manner similar to anaphase A, but instead

of kinetochores, the positions of poles were tracked in time with the first time frame determined as the frame when the distance between two poles started gradually increasing.

### Measuring the tilt of kinetochores at anaphase onset

Anaphase onset was determined in a frame just before the interkinetochore distance of the tracked kinetochore pair started gradually increasing. The coordinates of kinetochore pairs that ended up as errors and of error-free kinetochore pairs were tracked along with the coordinates of spindle poles. The angle that the kinetochores form with the long spindle axis was calculated using a home written Matlab script.

## Acknowledgements

We thank Alexey Khodjakov, Ina Poser and Tony Hyman, Thomas Surrey, Emanuele Roscioli, Andrew McAinsh, Mariola Chacon and Marin Barišić for the cell lines. We also thank Marko Šprem and Abberior team for help with developing microscopy protocols, Ivana Štimac and Marko Šprem for help with creating the MatLab and R scripts, Josip Čačković and Arian Ivec for technical assistance with the initial experiments and calculations, all members of Tolić and Pavin groups for discussions and advice, Ivana Šarić for the drawings. Funding This work was funded by the European Research Council (ERC Consolidator Grant, GA number 647077 and ERC Synergy Grant, GA number 855158), Croatian Science Foundation Cooperation Programme with Croatian Scientists in Diaspora "Research Cooperability" (HRZZ Project PZS-2019-02-7653), as well as QuantiXLie Centre of Excellence (Grant KK.01.1.1.01.0004) and IPSted (KK.01.1.1.04.0057) projects cofinanced by the Croatian Government and European Union through the European Regional Development Fund - the Competitiveness and Cohesion Operational Programme.

## Additional information

### Funding

| Funder | Grant reference number | Author |
| --- | --- | --- |
| European Research Council | ERC Consolidator Grant 647077 | Iva M Tolić |
| Croatian Science Foundation Cooperation Programme | HRZZ project PZS-2019-02-7653 | Iva M Tolić |
| European Regional Development Fund | QuantiXLie Centre of Excellence (KK.01.1.1.01.0004) | Iva M Tolić |
| European Research Council | ERC Synergy Grant 855158 | Iva M Tolić |
| European Regional Development Fund | IPSted (KK.01.1.1.04.0057) | Iva M Tolić |

The funders had no role in study design, data collection and interpretation, or the decision to submit the work for publication.

### Author contributions

Valentina Štimac, Isabella Koprivec, Conceptualization, Software, Formal analysis, Validation, Investigation, Visualization, Methodology, Writing – original draft; Martina Manenica, Formal analysis, Validation, Investigation, Visualization, Methodology; Juraj Simunić, Conceptualization, Supervision; Iva M Tolić, Conceptualization, Supervision, Funding acquisition, Writing – review and editing

### Author ORCIDs

Valentina Štimac http://orcid.org/0000-0003-0398-5493
Isabella Koprivec http://orcid.org/0000-0001-6486-8261
Iva M Tolić http://orcid.org/0000-0003-1305-7922

Decision letter and Author response
Decision letter https://doi.org/10.7554/eLife.83287.sa1
Author response https://doi.org/10.7554/eLife.83287.sa2

## Additional files

### Supplementary files
• MDAR checklist

### Data availability
All source codes and source data have been deposited to the Dryad repository (https://doi.org/10.5061/dryad.fn2z34tz7).

The following dataset was generated:

| Author(s) | Year | Dataset title | Dataset URL | Database and Identifier |
|---|---|---|---|---|
| Tolić IM | 2022 | Data from: Augmin prevents merotelic attachments by promoting proper arrangement of bridging and kinetochore fibers | https://dx.doi.org/10.5061/dryad.fn2z34tz7 | Dryad Digital Repository, 10.5061/dryad.fn2z34tz7 |

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
