## [Editor Report]

This manuscript investigates how the protein augmin contributes to correct spindle architecture and chromosome segregation during cell division in cultured human cells. High-quality super-resolution imaging and functional studies provide new insight into the mechanism of proper chromosome attachment to the spindle microtubules. This is an important paper that will be of interest to researchers interested in cell biology and cell biophysics.

---

## [Decision Letter]

[Editors' note: this paper was reviewed by Review Commons.]

---

## [Author Response]

*General Statements [optional]*

We thank the reviewers for providing thoughtful and constructive feedback on our manuscript. Motivated by their comments, we decided to perform a full revision with extensive new experiments, which led to important novel findings and thus a significant revision of the manuscript. The main changes lie in that we performed STED super resolution imaging of augmin depleted spindles, which allowed us to analyze the previously indistinguishable microtubule arrangements and erroneous attachments, as well as to quantify k-fibers and bridging fibers with unprecedented accuracy. Moreover, a key part of the revision is a functional approach where we tested how augmin depletion affects chromosome segregation fidelity, as suggested by Reviewer 3. Surprisingly, we found that without augmin, the lack of bridging fibers results in kinetochore tilt and, consequently, merotelic attachments in metaphase. Our STED images provide the first clear demonstration of such attachments, and these new findings offer a mechanistic explanation of how augmin-dependent microtubule branching promotes mitotic fidelity.

The main conceptual advance of our work is that the augmin complex is crucial for the proper organization of midplane-crossing microtubules into uniformly arranged bridging fibers that connect two sister k-fibers and extend nearly-parallel to the spindle axis. We found that bridging fibers are often absent following augmin-depletion, and have a specific role in the maintenance of interkinetochore tension, as well as in promoting mitotic fidelity by preventing merotelic attachments. K-fibers were also impaired, but to a lesser extent, with their thinning amplifying the negative effect on mitotic fidelity by compromising resolution of merotelic attachments later in anaphase. Altogether, we propose a model where augmin promotes mitotic fidelity by generating assemblies consisting of bridging and kinetochore fibers that align sister kinetochores to face opposite poles, thereby preventing erroneous attachments.

We have now addressed the criticism of the reviewers by identifying the specific effects of bridging fibers upon augmin depletion and considerably enhanced the significance by exploring the effects on chromosome segregation fidelity, thereby transforming our paper from a descriptive to a mechanistic study. As such, the study is no longer of interest only to the mitotic field, but to a larger scientific community including cell biologists, molecular biologists and biophysicists interested in microtubules, mitotic and meiotic spindles, cell division, chromosome segregation, aneuploidy, cancer, and development, as well as scientists developing quantitative super resolution imaging protocols for imaging of cellular structures.

Reviewer #1 (Evidence, reproducibility and clarity (Required)):In the manuscript "Augmin regulates kinetochore tension and spatial arrangement of spindle microtubules by nucleating bridging fibers", Manenica et al. explore the impact of augmin dependent microtubule nucleation on formation of a subset of spindle microtubules that bridge sister kinetochore fibers and investigate how this could affect the spindle forces and architecture. Using RNAi- and CRISPR-Cas9- based loss-of-function experimental approach, the authors propose that the bridging fibers are nucleated by augmin and that removal of augmin impairs proper spindle architecture, interkinetochore tension and microtubule poleward flux, specifically via its effect on the bridging fibers. Overall, the study is well designed and the manuscript well written. Expanding the knowledge on augmin contribution to the spindle functions and better understanding of the roles of bridging fibers would be important and of interest to cell biologists studying mitosis. Although this manuscript clearly shows that augmin depletion impairs the formation of bridging fibers (and other microtubules), the specific contribution of the bridging fibers to the augmin-dependent spindle functions is less clear.

We thank the reviewer for pointing out the quality of our experiment design and writing. We also appreciate the criticism regarding the exact contribution of bridging fibers to the spindle architecture, interkinetochore tension and microtubule poleward flux. We addressed this in our revised manuscript, where we studied the specific effect of bridging fibers on each of these phenotypes.

Using STED microscopy, we were now able to determine the specific effect of bridging fibers on spindle architecture by analyzing the previously indistinguishable microtubule arrangements within the crowded metaphase plate area and quantifying k-fibers and bridging fibers with unparalleled precision and accuracy. This type of analysis confirmed that augmin is vital for the organization of midplane-crossing microtubules into highly organized bridging fibers that connect sister kinetochore fibers and extend nearly-parallel to the spindle axis. Without augmin, proper bridging fibers were thin and often even absent, while the remaining midplane-crossing microtubules appear wavy and disorganized. K-fibers were also thinner, but to a lesser extent, as confirmed by two different types of analysis.

We clarified the contribution of bridging fibers to the interkinetochore tension using different types of analysis, which revealed that augmin-nucleated bridging microtubules have a specific role in the maintenance of interkinetochore tension. Indeed, the severe impairment of bridging fibers had a stronger effect on the interkinetochore distance than thinning of k-fibers following augmin depletion, and the specific effect of bridging fibers was demonstrated for kinetochores with the same k-fiber intensity.

As for poleward flux, a detailed analysis of poleward flux following augmin depletion, with a specific focus on the differences in bridging and k-fiber flux, is now published as a part of the recent Cell Reports paper from our lab (Risteski et al., 2022). Thus, this is no longer a main point of our manuscript, which now shifted its focus to more striking consequences of augmin depletion, particularly microtubule arrangements within the metaphase plate and mitotic fidelity.

The effect of augmin on mitotic fidelity, as suggested by Reviewer 3, was studied using sophisticated live-cell imaging and tracking protocols. We described the main results in the General Statement (see above), and dedicated an entire new section of the manuscript (pages 6-11 along with a new Figure 2) to these exciting new results that provide a mechanistic explanation of how augmin-dependent microtubule nucleation prevents merotelic attachments.

Major comments:1) Using cold treatment-induced microtubule destabilization, Zhu et al. (JCB 2008) showed that augmin depletion affected exclusively kinetochore microtubules. Since the bridging microtubules are usually not visible in the cold-treated spindles (due to being less stable/cold resistant compared to the k-fibers), it is unlikely that the observed effects were mainly associated with the bridging fibers. Thus, it would be important to further clarify the respective contribution of augmin to the formation of k-fibers and the bridging fibers. The cold-treatment experiment performed by Zhu et al. could be used in RPE1 and HeLa-PRC1-GFP cells to address the contribution of augmin nucleation to kinetochore- vs. bridging microtubules from another angle.Because of the above mentioned results by Zhu et al. it is difficult to grasp how augmin depletion could have a bigger effect on the bridging fibers than on the k-fibers, as concluded from the Figure 2C data. In fact, Figure 2A clearly shows a strong effect on k-fibers in spindles where the bridging fibers are reduced/missing.Also, Figure 1 D and E suggest that HAUS8 siRNA exclusively affected the bridging fibers, leaving the k-fibers intact, which is again against the data reported in Zhu et al. 2008 and in contrast with the representing image shown in Figure 1B. Even if the RNAi was less efficient compared to HAUS6 RNAi, as the authors proposed, this could still not explain the observed discrepancy.

We thank the reviewer for this comment. To better address the contribution of augmin to either bridging or kinetochore microtubule nucleation, we repeated the experiments by using STED super-resolution microscopy on cells immunostained for tubulin, with or without HAUS6/8 depletion. STED imaging enabled a previously unattainable and exceptionally detailed visualization of k-fibers and bridging fibers (Figure 1 and Figure 3A), and subsequent analysis confirmed our initial result that augmin depletion has a larger effect on bridging fibers than on kfibers (Figure 3D,E). Importantly, due to the increased precision of our measurements when using STED images, the new results for HAUS6 siRNA and HAUS8 siRNA were now similar, and showed that the lack of augmin affects k-fibers to an extent comparable to Zhu et al., 2008 (Figure 3D,E). The data from the original manuscript, which were obtained by confocal microscopy and were therefore somewhat less precise due to the intrinsic limitations of the method, are now shown in Supplementary Figure S3.

To independently test the effect of augmin depletion on k-fibers, we also performed experiments with cold treatment as suggested, imaged the cells by STED microscopy, and analyzed the images as in Zhu et al., 2008 (Figure 3F,G). These experiments were done on RPE1 cells that stably express CENP-A-GFP. Cold treatment in HeLa-PRC1-GFP cells was no longer necessary, as STED imaging clearly showed the absence of bridging fibers in cold treated cells. The new results, that emphasise the effect on bridging fibers but no longer state that k-fibers are intact are described on page 12:

“Quantification of STED images further revealed that HAUS6 depletion resulted in 68 ± 8% reduction of the bridging fiber signal intensity and 24 ± 6% reduction of the k-fiber signal intensity, with similar results obtained by HAUS8 depletion (Figure 3D-E). These data indicate that augmin depletion affects not only k-fibers, but even more so bridging fibers. The contribution of augmin to the nucleation of k-fibers was independently tested by measuring their intensity in spindles exposed to cold treatment in which bridging fibers are removed (Figure 3F). HAUS6 depletion resulted in a 37 ± 5% reduction of the k-fibers (Figure 3G), which is consistent with a previous study (Zhu et al., 2008) and comparable to values under non-cold conditions.”

2) The authors showed that kinetochore pairs in the outer parts of Augmin-depleted spindles have larger inter-kinetochore distance compared to those in the inner parts of spindles. They indirectly related this to a predominant presence of the bridging fibers in the outer parts, concluding that augmin regulates inter-kinetochore tension via nucleation of the bridging fibers. A more direct way would be to show the eventual positive correlation between the inter-kinetochore distance and the bridging- and k- fibers intensity. Also, it would be nice to include the quantifications and correlation data for inter-kinetochore distance, distance from the spindle axis and the bridging- and k- fibers intensities for the control cells too.

We analyzed the new data and included the correlations in the altered manuscript. We explained the correlation data for the interkinetochore distance and the distance from the spindle axis as follows: “… we noticed that the interkinetochore distance was smaller in the inner part of the spindle in augmin-depleted cells (Figure 5A-D, Supplementary Figure S5B), where bridging fibers were most severely impaired (Figure 3H and 4A). This was not the case in control cells, which showed no difference in interkinetochore distance between the inner and the outer part of the spindle (Figure 5D, Supplementary Figure S5B).”

We also included the correlation data for the interkinetochore distance and the bridging and kfibers intensity: “… we found that although the interkinetochore distance correlated both with bridging and k-fiber intensity after augmin depletion, the correlation with bridging fiber intensity was stronger (Supplementary Figure S5D-E). Such correlations were absent in control cells (Supplementary Figure S5D-E).”

In addition, we performed yet another type of analysis where we divided kinetochore pairs into subsets with either comparable bridging fiber intensities and contrasting k-fiber intensities or comparable k-fiber intensities and contrasting bridging fiber intensities. We then compared the interkinetochore distance of kinetochore pairs within these subsets, as follows: "To explore a specific contribution of k-fibers to the interkinetochore tension, we divided the kinetochore pairs in augmin-depleted cells into two subsets, those with and without bridging fibers, and found that the interkinetochore distance did not correlate with k-fiber intensity within each group (Figure 5F), which argues against the k-fiber intensity as a sole determinant of interkinetochore tension. In agreement with this, when we selected two subsets of kinetochore pairs with either very strong or very weak k-fiber intensity but with comparable bridging fiber intensities (Supplementary Figure S5F-G), we found no difference in the interkinetochore distance between these subsets (Supplementary Figure S5H). Finally, to examine a specific contribution of bridging fibers, we identified two subsets of kinetochore pairs with similar k-fiber intensity values, one of which had bridging fibers and the other which did not (Supplementary Figure S5I). We found that the interkinetochore distance was larger in the subset with bridging fibers than without (Figure 5G), indicating a specific effect of bridging fibers on interkinetochore tension. Analysis of live-cell confocal images of RPE1 cells yielded similar results (Supplementary Figure S5J-M). Based on these data, we conclude that augmin has a significant role in regulating interkinetochore tension through the nucleation of bridging microtubules."

In sum, in the altered manuscript we conclude that both bridging and k-fibers regulate the kinetochore tension, and that in augmin-depleted cells there is indeed a specific effect of bridging fibers on interkinetochore tension, which is larger than the effect of k-fibers.

3) It is stated in the manuscript that the k-fibers without bridging fibers have shorter contour length compared to the k-fibers with bridging fibers, and that the curvature of k-fibers lacking the bridging fibers is drastically reduced. However, the data in Figure 5D and Table 1 show a slight effect on the contour length of the k-fibers lacking the bridging fibers compared to the ones containing the bridging fibers only in RPE1 siHAUS8 cells, while this effect seems to be missing in RPE1 HAUS8 KO cells, as well as in siHAUS6 in RPE1 and HeLa cells.Figure 2 shows that the kinetochore pairs without the bridging fibers are located closer to the spindle axis. Thus, it is not clear whether the effect on curvature observed in the augmin depleted cells is independent of the position of kinetochore pairs within the spindle, as the spindle axis-proximal pairs would anyway have a bigger radius compared to the more distant ones.

As these analyses were previously performed on bundles stained with SiR-tubulin and using confocal microscopy, we have now determined their curvature on spindles immunostained for tubulin and imaged by STED microscopy, where the shape of these bundles can be determined more precisely both in control and HAUS6 depleted cells. In the revised manuscript, only those spindles were taken for further analysis (Supplementary Figure S3I-K), as this is no longer a major point in the revised manuscript which shifted its focus towards other more striking consequences of augmin depletion. The text was revised as follows: “Whereas the bundles without kinetochores in HAUS6 siRNA-treated cells had a significantly longer contour when compared to all other bundle types (Supplementary Figure S3J), k-fibers without bridging fibers in augmin-depleted cells had a significantly larger radius of curvature than any of the other bundle types in augmin-depleted or control cells (Supplementary Figure S3K). Taken together, the outer interpolar bundles without associated kinetochores are excessively long and make the spindle wider, whereas k-fibers lacking a bridging fiber are overly straight, ultimately resulting in a diamond-like shape of the spindle.”

As for previous Figure 2, in all experiments regarding shape, we only analyzed the outermost bundles, so the potential effect of the position of kinetochore pairs within the spindle can be excluded. We explained that in the Methods section and highlighted it in the caption of the Supplementary Figure 3I: “In control cells, only the outermost bundle was tracked. In HAUS6 siRNA treated cells, three different groups of outermost bundles were tracked: bundles with visible bridging fibers, bundles with no visible bridging fibers and curved bundles extending far from the metaphase plate” and “Examples of each bundle type are shown in insets. From left to right: the outermost bundle in control cells, the outermost bundle with a bridging fiber, the outermost bundle without a bridging fiber and the outermost bundle without kinetochores in HAUS6-depleted cells”, respectively.

4) The authors reported that augmin depletion impairs microtubule poleward flux and conclude that this happens exclusively due to the perturbation of bridging fibers. While the results from this and other studies clearly show that augmin depletion perturbs spindle microtubules in general, it is not clear whether this had a stronger effect on the bridging microtubules (see the comments in point 1). Thus, the impact of augmin depletion on kinetochore microtubules or other antiparallel microtubules within the spindle (e.g. the ones recently shown in O’Toole et al., MboC 2020) cannot be ruled out as a potential cause of the impaired microtubule flux. Also, Steblyanko et al. (EMBO J, 2020) showed that PRC1 depletion hadffectt on microtubule poleward flux in metaphase cells. Since it has been previously shown by the authors of this manuscript that PRC1 depletion disrupts the formation of bridging fibers, it is unlikely that the bridging fibers are the main cause of the augmin depletion-mediated effect on the microtubule flux.

We modified the text on poleward flux to include the contribution of both bridging and k-fibers. We also performed new experiments on U2OS cells and included references to the new work from our lab recently published in Cell Reports (Risteski et al., 2022), which was able to distinguish between the effect of augmin depletion on bridging and k-fiber flux. The effect of augmin on poleward flux is now only a minor point in our revised manuscript which mainly focuses on microtubule arrangements and mitotic fidelity.

However, as requested, we included a comment on PRC1 depletion: “Recent speckle microscopy experiments in RPE1 cells, which were able to separate the effect of augmin on poleward flux of bridging and k-fibers, revealed that both k-fibers and the remaining bridging fibers were significantly slowed down (Risteski et al., 2022). Bridging fibers fluxed faster than k-fibers in control and augmin-depleted cells (Risteski et al., 2022), supporting the model in which poleward flux is largely driven by sliding apart of antiparallel microtubules (Brust-Mascher et al., 2009; Mitchison, 2005; Miyamoto et al., 2004). We propose that augmin depletion results in slower flux of bridging fibers because the remaining bridging microtubules are likely nucleated at the poles, where microtubule depolymerization mechanisms might curb poleward flux speed (Ganem et al., 2005). In contrast, PRC1 depletion does not affect the flux (Risteski et al., 2021 Preprint; Steblyanko et al., 2020) even though it reduces bridging fibers (Kajtez et al., 2016; Polak et al., 2017), possibly because the remaining bridging microtubules are generated away from the poles via augmin and can thus flux freely.”

Minor comments:1) Introduction: chromatin- and kinetochore- mediated generation of spindle microtubules are ignored when describing the origins of spindle microtubules in human somatic cells.

We included the chromatin- and kinetochore-mediated generation of spindle microtubules in the Introduction. We revised the text as follows: “Spindle microtubules in human somatic cells are generated by several nucleation mechanisms, including centrosome-dependent and augmin dependent nucleation (Kirschner and Mitchison, 1986; Pavin and Tolić, 2016; Petry, 2016; Prosser and Pelletier, 2017; Wu et al., 2008; Zhu et al., 2008), with an addition of chromatin- and kinetochore-dependent nucleation as a third mechanism that contributes to the directional formation of k-fibers (Maiato et al., 2004; Sikirzhytski et al., 2018; Tulu et al., 2006).”

2) The authors proposed less efficient HAUS8 depletion as a potential reason of discrepancy between the siHAUS6 and siHAUS8 results. This should be shown by Western blot, like it is presented for the RNAi efficiency of siHAUS6.

We agree with the reviewer that it would be best to include Western blot for the RNAi efficiency of siHAUS8. However, as we explained in the Methods section, commercially available HAUS8 antibodies resulted in no detectable bands in our hands, regardless of the modifications in the Western blot protocol. We explained this in the Methods section, as follows: “Rabbit polyclonal HAUS8 antibody (diluted 1:1000, PA5-21331, Invitrogen and NBP2-42849, Novus Biologicals) resulted in no detectable bands under these conditions”. For this reason, we performed immunocytochemistry to determine the efficiency of siHAUS8. The discrepancy was now also addressed as a part of our new super resolution STED analysis, which enabled increased precision and where depletion of HAUS6 and HAUS8 produced the same results.

3) The measurements of total PRC1 intensities are mentioned in the manuscript text, but not shown in the figures.

PRC1 measurements are now performed on both RPE1 and HeLa cells with corresponding graphs shown in Figure 4C and Supplementary Figure S4C.

4) Supplementary Videos 3 and 4 are wrongly annotated as Supplementary Videos 1 and 2 in the text.

As we have a new set of videos, this is no longer applicable.

5) Given the spindle length phenotypes are opposite in HeLa and RPE1 cells, in order to be consistent with the other experiments it would be better to perform the PRC1 measurements in RPE1 cells (e.g. using the anti-PRC1 antibody as shown in Supplementary Figure 3B).

We have now performed size measurements in all three cell lines: RPE1 cells stably expressing CENP-A-GFP and Centrin1-GFP, RPE1 cells stably expressing PRC1-GFP and HeLa cells stably expressing PRC1-GFP treated with MG-132. These results are now shown in Supplementary Figure S4J-K. The phenotypes remained the same as in the original experiments. We revised the text to better explain the observed differences as follows: “While the spindles in RPE1 cells shortened following augmin depletion, those in HeLa cells were longer (Supplementary Figure S4J), consistent with previous observations on *Drosophila* S2 cells and *Xenopus* egg extracts (Goshima et al., 2007; Petry et al., 2011). This difference in spindle length might be due to the overlaps remaining the same length after augmin depletion in RPE1 cells, while being longer and thereby able to push the spindle poles further apart in HeLa cells (Supplementary Figure S4K).”

Please note that due to the extensive amount of new data regarding microtubule arrangements and segregation errors obtained from STED microscopy and live-cell imaging of kinetochores over time, size measurements are now only a small part of our revised story and are, as such, no longer presented in the main figure.

6) Why are the microtubule flux rates for RPE1-PA-GFP-α-tubulin cells measured in this study largely different than the rates reported for the same cell line in Dudka et al., Nat Comms 2018 and Dudka et al., Curr Biol 2019? In order to better understand this difference and strengthen the microtubule flux data, it would be helpful to increase the experimental numbers to match the ones used in the mentioned studies.

We performed photoactivation experiments on a higher number of U2OS cells stably expressing CENP-A GFP, mCherry-tubulin and PA-tubulin (N = 30 measured photoactivation spots in 30 control and HAUS6-depleted cells, see Supplementary Figure S3L-M). U2OS cells with labelled kinetochores and tubulin were used to exclude the potential effects of SiR-tubulin on poleward flux, as well as to better determine the position of the metaphase plate. The results in control cells are comparable to the poleward flux measured in the same cell line (Steblyanko et al., 2020).

Similar to shape and size measurements, due to the extensive and exciting new data regarding microtubule arrangements and segregation errors and because poleward flux was studied into much more detail in a recent Cell Reports study from our lab, this is no longer a major point in our revised manuscript. Poleward flux, along with shape analysis, is now included in the Supplementary Figure 3, as a part of the section about spindle architecture and dynamics.

7) The number of cells used per each experiment should be clearly stated.

In all experiments included in the main figures, we have now performed 3 independent experiments with at least 10 cells each. The numbers are also clearly stated in the captions of figures for all experiments.

Reviewer #1 (Significance (Required)):This study expands the analysis of augmin contribution to the spindle functions and focuses on its role in formation of the bridging fibers, which is of interest to cell biologists studying mitosis. It clearly shows that in addition to its effect on the k-fibers, augmin depletion also impairs the formation of bridging fibers. However, the exact contribution of the bridging fibers to the spindle functions affected by augmin depletion remains unclear.

We thank the reviewer for the thoughtful comments and are convinced that our new experiments which include the never-before-seen arrangements of microtubules, along with functional analysis of augmin-depleted cells, demonstrated that bridging fibers have a specific k-fiber independent effect on kinetochore tension and a specific role in preventing merotelic attachments.

Since our revised manuscript includes both the first quantitative analysis of the human spindle using STED microscopy, as well as new mechanisms by which augmin ensures mitotic fidelity, it is no longer of interest only to cell biologists studying mitosis but also to a wider scientific community, including cell biologists, molecular biologists and biophysicists interested in microtubules, mitotic and meiotic spindles, cell division, chromosome segregation, aneuploidy, cancer, and development, as well as scientists developing quantitative superresolution imaging protocols for imaging of cellular structures.

Reviewer #2 (Evidence, reproducibility and clarity (Required)):Summary:The authors found that the microtubules in the bridging fibres of the mitotic spindle in a human cell line are predominantly supplied via augmin-dependent nucleation. On the other hand, the contribution of augmin to kinetochore fibre formation is ~40%. Augmin-depleted cells showed reduced inter-kinetochore tension and slower poleward flux of spindle microtubules, suggesting that bridging fibres play a role in these events. This study expands our knowledge on the role of augmin and augmin-mediated microtubules in animal somatic cells.

We thank the reviewer for this accurate summary of our work. In addition to the findings nicely described here, we now have an entire new section on the role of augmin in mitotic fidelity, as suggested by Reviewer 3 and presented in the new Figure 2.

Major comments:– Are the key conclusions convincing?Yes.– Should the authors qualify some of their claims as preliminary or speculative, or remove them altogether?In the current manuscript, the slower flux is attributed solely to the lack of bridging fibres in the augmin-depleted cells. This is an overinterpretation, as the augmin's role in the spindle is not limited to generating bridging fibres.

We modified the text on poleward flux to include the contribution of both bridging and k-fibers. We also performed new experiments on U2OS cells and included references to the new work from our lab recently published in Cell Reports (Risteski et al., 2022), which was able to distinguish between the effect of augmin depletion on bridging and k-fiber flux. Because poleward flux was analyzed into much more detail in this study, and because we produced surprising new data regarding microtubule arrangements and segregation errors, poleward flux is no longer a major point in our revised manuscript. It is now included in the Supplementary Figure 3 along with shape analysis, as a part of the section about spindle architecture and dynamics.

The text about flux was modified as follows: “Recent speckle microscopy experiments in RPE1 cells, which were able to separate the effect of augmin on poleward flux of bridging and k-fibers, revealed that both k-fibers and the remaining bridging fibers were significantly slowed down (Risteski et al., 2022). Bridging fibers fluxed faster than k-fibers in control and augmin-depleted cells (Risteski et al., 2022), supporting the model in which poleward flux is largely driven by sliding apart of antiparallel microtubules (Brust-Mascher et al., 2009; Mitchison, 2005; Miyamoto et al., 2004). We propose that augmin depletion results in slower flux of bridging fibers because the remaining bridging microtubules are likely nucleated at the poles, where microtubule depolymerization mechanisms might curb poleward flux speed (Ganem et al., 2005). In contrast, PRC1 depletion does not affect the flux (Risteski et al., 2021 Preprint; Steblyanko et al., 2020) even though it reduces bridging fibers (Kajtez et al., 2016; Polak et al., 2017), possibly because the remaining bridging microtubules are generated away from the poles via augmin and can thus flux freely.”

– Would additional experiments be essential to support the claims of the paper? Request additional experiments only where necessary for the paper as it is, and do not ask authors to open new lines of experimentation.No.– Are the suggested experiments realistic in terms of time and resources? It would help if you could add an estimated cost and time investment for substantial experiments.N/A– Are the data and the methods presented in such a way that they can be reproducedYes.– Are the experiments adequately replicated and statistical analysis adequate?Yes.Minor comments:– Specific experimental issues that are easily addressable.None.– Are prior studies referenced appropriately?Yes.– Are the text and figures clear and accurate?1) Page 15: "To determine the curvature of the bundles, we ….. with all other bundles types (Figure 5E)." – I could not understand this sentence well, and would like to ask for a revision.

The text has now been changed to: “To gain insight into the contribution of each of these functionally distinct microtubule bundles to the maintenance of spindle geometry, we traced the outermost bundles in HAUS6 siRNA treated RPE1 cells imaged using STED microscopy and fitted a circle to the bundle outline (Supplementary Figure S3I, see Methods)”.

2) The following words may be too strong:Page 20: whereas k-fiber microtubules are "mainly" nucleated in an augmin-independent manner (could 61% contribution be called "mainly?").

We revised this sentence on page 24 as follows: “K-fibers were also thinner, though to a lesser extent, indicating that they are largely nucleated in an augmin-independent manner, at the centrosome or kinetochores and chromosomes.”

Page 21, bottom: "demonstrates".

As we changed this section of the manuscript, this is no longer applicable.

– Do you have suggestions that would help the authors improve the presentation of their data and conclusions?No.Reviewer #2 (Significance (Required)):– Describe the nature and significance of the advance (e.g. conceptual, technical, clinical) for the field.The presence of bridging fibres has been recognised for decades; however, until recently, little attention has been paid to this structure from a mechanistic and functional point of view. The Tolic lab has been shedding light on this structure for the past several years. The current study represents another step forward in the research of the origin and function of bridging fibres.– Place the work in the context of the existing literature (provide references, where appropriate).Augmin's critical contribution to microtubule nucleation in the human somatic spindle has been well documented, as cited by the authors. The current study is the first to show that augmin also contributes to bridging fibres. The >70% contribution may be more than expected, given that centrosomal microtubules frequently reach the spindle midzone.Reduced inter-kinetochore tension has also been documented, but previous studies attributed this exclusively to reduced number of kinetochore microtubules. The current study has revised this view.

We thank the reviewer for correctly pointing out the main points of our original manuscript. These results are now strengthened by new experiments, and accompanied by STED microscopy imaging and a section about the role of augmin-nucleated microtubules in the maintenance of mitotic fidelity.

– State what audience might be interested in and influenced by the reported findings. Spindle researchers.

Since our revised manuscript includes both the first quantitative analysis of the human spindle using STED microscopy, as well as new mechanisms by which augmin ensures mitotic fidelity, it is no longer of interest only to spindle researchers but also to a wider scientific community, including cell biologists, molecular biologists and biophysicists interested in microtubules, mitotic and meiotic spindles, cell division, chromosome segregation, aneuploidy, cancer, and development, as well as scientists developing quantitative super resolution imaging protocols for imaging of cellular structures.

– Define your field of expertise with a few keywords to help the authors contextualize your point of view. Indicate if there are any parts of the paper that you do not have sufficient expertise to evaluate.This review is written by a researcher who is familiar with the literature of the mitotic spindle.

We thank the reviewer for an accurate summary of our work and perceptive comments.

Reviewer #3 (Evidence, reproducibility and clarity (Required)):In this study Manenica et study how the presence of the augmin complex affects the overall spindle architecture and the different types of spindle microtubules. The authors propose that depletion of augmin affects particularly bridging microtubules, leading to their disappearance on sister-kinetochores located in the central part of the metaphase plate.Overall the manuscript is well written, clear and supported by excellent explanatory schemes. The main conclusion of the manuscript, i.e. that augmin plays an essential role in the formation of bridging microtubules is generally well supported by the data. A number of other conclusions, however, are less well supported by the data and would benefit from a number of additional experiments, repetitions or analysis. Specifically:

We thank the reviewer for complimenting the clarity of our manuscript, and the data supporting the main conclusion. Other conclusions, which the reviewer states were not equally well supported by the data, are now strengthened by a new set of experiments using STED microscopy, increased repetitions (at least 30 cells from 3 independent experiments) and multiple new analyses. In addition, our revised manuscript now also includes both the first quantitative analysis of the human spindle using STED microscopy, as well as an entire new section on the mechanisms by which augmin ensures mitotic fidelity, as thoughtfully suggested below.

1) Throughout all the figures the authors use a t-test, which is fine when comparing two conditions, but not for multiple experimental conditions. The authors should instead use an ANOVA test or apply a Bonferroni correction. This can strongly affect the significance of some of the reported results.

We agree and have performed either ANOVA with post-hoc Tukey test or Mann-Whitney U-test instead of t-test where appropriate throughout the whole manuscript. The statistical analyses that were used are clearly stated in the captions of the figures.

2) Another general concern is that the authors rely throughout the manuscript on live cell imaging data from few cells (5-10). Live cell imaging data has the advantage to avoid fixation artifacts, but the low sample size is a major concern, as for every experiment the authors rely on 10 cells (and for inter-kinetochore distances on 5) in three independent experiments overall. This means that two our of three of those independent experiments are based on 3 cells only, which is too low, given that siRNA depletions are known to be variable in their efficiencies. With such low number of cells, there is always the danger of an unconscious selection bias, which can skew a result. Just to take an example the spindle length, structure and density for HAUS-8 depleted RPE1 cells looks very different in the examples show in Figure 1B, 2A, or 5B.It is therefore essential to work with a higher sample size, at minimum 10 cells per independent experiment.

This is a valid critique, which we addressed by performing 3 independent experiments with at least 10 cells each for all of our new analyses included in the main figures.

3) Throughout all experiments the authors use 100nm Sir-Tubulin, which in our hands already leads to substantial changes in microtubule dynamics, as it stabilizes spindle microtubules. I understand why the authors did this, as they wanted to also stain for weak bridging fibers, but tt would be important to validate some of the obtained results with an independent approach, for example fixed-cell imaging and tubulin staining, to rule out artifacts introduced by SiR-tubulin.

We thank the reviewer for this suggestion. To validate our results, we performed tubulin immunostaining, as suggested. Moreover, we imaged the immunostained cells by using super resolution STED microscopy, which significantly increased the precision of our measurements, and included these results in the main figures (Figure 1, Figure 2J and K, Figure 3, Figure 5). The data obtained from cells stained with SiR-tubulin and imaged using live-cell confocal microscopy are now shown in Supplementary figures.

4) Figure 1: Given that the authors later report that augmin affects more strongly bridging fibers in the central part of the spindle, how were the values in terms of microtubule densities obtained in the experiments in Figure 1: only on the outer microtubules, or overall in the spindle?

Values in Figure 1 (now Figure 3) were obtained overall in the spindle. We described the selection in the text as follows: “We measured tubulin signal intensity of randomly selected bridging (Ib) and k-fibers (Ik) which had no other microtubules in their immediate neighborhood…”

5) Figure 2: the authors conclude that depletion of augmin has a much stronger effect on the bridging fibers located in the central part of the spindle. This is a very interesting result, but it begs the question as to the origin of this difference. If the authors analyze in control cells the interkinetochore distances and the density of the bridging fibers of the kinetochores located in the central part of the metaphase plate vs those located at the outer part of the plate, do they already see a difference? In other words, is the effect of augmin due to already weaker bridging fibers in the central part of the spindle, or is the depletion effect indeed specific for those bridging fibers located in the middle. This analysis should be possible with the existing data (+ a higher sample size)

This was now analyzed in the cells imaged using STED microscopy with increased resolution and a higher sample size. From our new data, it is clear that the depletion effect is indeed specific for bridging fibers located in the middle as there was no significant difference between the interkinetochore distance in the inner and the outer part of the spindles in control cells (Figure 5D and Supplementary Figure S5B). The same trend can also be seen for bridging fiber density (Figure 3H). We modified the text as follows: “However, we noticed that the interkinetochore distance was smaller in the inner part of the spindle in augmin-depleted cells (Figure 5A-D, Supplementary Figure S5B), where bridging fibers were most severely impaired (Figure 3H and 4A). This was not the case in control cells, which showed no difference in interkinetochore distance between the inner and the outer part of the spindle (Figure 5D, Supplementary Figure S5B).”

6) Figure 4: the authors study spindle width, length and diameter of the metaphase plate in a small number of cells (10). One concern is that these values might change as cells progress from late prometaphase to anaphase onset (metaphase plate width decreases for example). Given the low number of cells the authors do not know if they are comparing cells at similar mitotic times. To circumvent this issues, they could: either arrest the cells with MG132 for 1 hour, to obtain an endpoint, or record these different values as cells progress through mitosis and thus be able to compare similar conditions.

We agree with this suggestion, and we performed new experiments by arresting the cells with MG-132: “… in HeLa (Kajtez et al., 2016) and RPE1 (Asthana et al., 2021) cells stably expressing PRC1-GFP with and without MG-132 treatment (Figure 4A-B, Supplementary Figure S4A).” We measured spindle width, length and diameter of the metaphase plate in arrested RPE1 cells stably expressing CENP-A-GFP and Centrin1-GFP, RPE1 cells stably expressing PRC1-GFP, and HeLa cells stably expressing PRC1-GFP. We treated the cells with MG-132 for 30 minutes, as this was in our hands enough to arrest the cells, without causing other changes, e.g., problems with spindle orientation that occur after 1 hour of treatment. The results are now part of the Supplementary Figure S4 and are obtained from three independent experiments with at least 10 cells per experiment.

7) In the discussion the authors conclude that the longer bundles and the reduction in microtubule poleward flux is due to the absence of bridging microtubules. This is an over-interpretation as augmin could in theory affect these parameters independently of the bridging microtubules, longer bundles could be generally due to the reduced number of microtubules in the k-fibers and the bridging microtubules. A better control would be to affect bridging microtubules with an independent tool, such as PRC1 depletion, and to measure these paramenters in the same RPE1 cell line, since differences can arise from cell line to cell line as the authors also document in their study (for example spindle length in Figure 4).

We modified the Discussion based on new results, so these statements are now in the Results section. For the long, curved bundles, we modified the sentence as follows: “These bundles likely arose either due to PRC1 crosslinking excessively long astral microtubules that were now able to reach the spindle midzone or due to PRC1 activity combined with the excess of free tubulin present as a consequence of less tubulin being incorporated in bridging and k-fibers.”

Regarding the reduced poleward flux following augmin depletion, we revised the text as follows: “Recent speckle microscopy experiments in RPE1 cells, which were able to separate the effect of augmin on poleward flux of bridging and k-fibers, revealed that both k-fibers and the remaining bridging fibers were significantly slowed down (Risteski et al., 2021 Preprint). Bridging fibers fluxed faster than k-fibers in control and augmin-depleted cells (Risteski et al., 2021 Preprint), supporting the model in which poleward flux is largely driven by sliding apart of antiparallel microtubules (Brust-Mascher et al., 2009; Mitchison, 2005; Miyamoto et al., 2004). We propose that augmin depletion results in slower flux of bridging fibers because the remaining bridging microtubules are likely nucleated at the poles, where microtubule depolymerization mechanisms might curb poleward flux speed (Ganem et al., 2005). In contrast, PRC1 depletion does not affect the flux (Risteski et al., 2021 Preprint; Steblyanko et al., 2020) even though it reduces bridging fibers (Kajtez et al., 2016; Polak et al., 2017), possibly because the remaining bridging microtubules are generated away from the poles via augmin and can thus flux freely.”

However, due to the extensive and thrilling new data regarding microtubule arrangements and segregation errors and because poleward flux was studied into much more detail in a recent Cell Reports study from our lab (Risteski et al., 2022), this is no longer a major point in our revised manuscript. Poleward flux, along with shape analysis, is now included in the Supplementary Figure 3, as a part of the section about spindle architecture and dynamics.

Minor comment:– The reported flux rate for control-depleted cells is substantially higher than the flux rates normally reported for human cells. This could be due to the experimental conditions (slight changes in temperature), but at minimum the authors should comment on this.

We performed photoactivation experiments on a higher number of U2OS cells stably expressing CENP-A GFP, mCherry-tubulin and PA-tubulin (N = 30 measured photoactivation spots in 30 control and HAUS6-depleted cells, see Supplementary Figure S3L-M). U2OS cells with labelled kinetochores and tubulin were used to exclude the potential effects of SiR-tubulin on poleward flux, as well as to better determine the position of the metaphase plate. The results in control cells are comparable to the poleward flux measured in the same cell line (Steblyanko et al., 2020).

Please note that, as previously stated, this is no longer a major point in our revised manuscript and can be found in Supplementary Figure 3.

Reviewer #3 (Significance (Required)):The significance of the study is that the authors performed a detailed description of the effects of augmin depletion on the spindle architecture, in particular bridging fibers. Nevertheless, many of the reported results are already known (and as cited by the authors): the reduction in interkinetochore distances or the change in spindle architecture. The 3 main novel results, is the fact that augmin affects more bridging microtubules, particularly in the central part of the spindle, and that it also affect poleward microtubule flux, which limits the impact of this study to a specialized mitotic spindle audience. Nevertheless, if the authors address the reviewers concerns, this could be a nice, descriptive study for the mitotic field.One way to expand the significance of this study would be to test how augmin depletion and the lack of bridging microtubules in the central part of the metaphase plate affects chromosome segregation. Does the specific absence of bridges in this part lead to more lagging chromosomes, chromosome segregation errors, or micronuclei amongst sister chromatids located in the central part of the spindle? Is there a differential anaphase A speed for those kinetochore vs those at the periphery that still are associated to bridging fibers? Such a functional approach could allow to highlight the most interesting aspect of this study, the spatial difference in the effects of augmin depletion. Such experiments would, however, not be part of a revision, but rather a substantial enhancement of the present study.Patrick Meraldi

This is a great idea that we truly appreciate! We performed new experiments to study lagging chromosomes and indeed found that they were more often located in the inner part of the spindle in HAUS6-depleted than in control cells, which is likely due to the specific impairment of bridging fibers in that area. We also found that lagging chromosomes typically had a lower interkinetochore distance and a higher kinetochore tilt just before the onset of anaphase, which we interpret as a signature of perturbed bridging fibers. Using STED microscopy, we were able to clearly see merotelic attachments at kinetochores lacking a bridging fiber in metaphase following augmin depletion. Thus, we propose that augmin-nucleated bridging fibers prevent merotelic attachments by creating a nearly parallel and highly bundled microtubule arrangement unfavorable for creating additional attachments. On the other hand, augmin-nucleated k-fibers produce robust force required to resolve errors during anaphase. We dedicated an entire new section on pages 6-11 and a new Figure 2 to these exciting new results.

We agree with the Reviewer 3 that these results substantially increase the significance of this study, which is now no longer descriptive nor of interest only to the mitotic field, but offers a new mechanism that safeguards mitotic fidelity, which makes it of interest to a wider scientific community. This includes cell biologists, molecular biologists and biophysicists interested in microtubules, mitotic and meiotic spindles, cell division, chromosome segregation, aneuploidy, cancer, and development, as well as scientists developing quantitative superresolution imaging protocols for imaging of cellular structures.

We are thankful to the reviewer for raising such an interesting question, and motivating us to use a functional approach to study the mechanism by which the observed effects of augmin depletion compromise mitotic fidelity.